# Strain-induced long-range charge-density wave order in the optimally doped Bi$_2$Sr$_{2−x}$La$_x$CuO$_6$ superconductor

Shinji Kawasaki [1], Nao Tsukuda[1], Chengtian Lin[2] & Guo-qing Zheng [1]

The mechanism of high-temperature superconductivity in copper oxides (cuprate) remains elusive, with the pseudogap phase considered a potential factor. Recent attention has focused on a long-range symmetry-broken charge-density wave (CDW) order in the underdoped regime, induced by strong magnetic fields. Here by $^{63,65}$Cu-nuclear magnetic resonance, we report the discovery of a long-range CDW order in the optimally doped Bi$_2$Sr$_{2−x}$La$_x$CuO$_6$ superconductor, induced by in-plane strain exceeding $|\varepsilon| = 0.15$ %, which deliberately breaks the crystal symmetry of the CuO$_2$ plane. We find that compressive/tensile strains reduce superconductivity but enhance CDW, leaving superconductivity to coexist with CDW. The findings show that a long-range CDW order is an underlying hidden order in the pseudogap state, not limited to the underdoped regime, becoming apparent under strain. Our result sheds light on the intertwining of various orders in the cuprates.

Strongly correlated electron systems exhibit symmetry-broken states, such as spin and charge-ordered states[1]. Unconventional superconductivity arises inside or near these states, likely due to quantum fluctuations of the orders[2–4]. Therefore, to understand the mechanism of unconventional superconductivity, it is essential to reveal the nature of the underlying symmetry-broken states[5]. There are two well-known examples of background electronic states that are often discussed as having broken symmetry.

The first one is the U-based *5f*-electron heavy fermion superconductor URu$_2$Si$_2$[6,7], where a mysterious phase transition occurs at $T_{HO} = 17.5$ K with a hidden order, followed by unconventional *d*-wave superconductivity with transition temperature $T_c = 1.5$ K[6,7]. Despite several decades of research from 1985, the order parameter has not been determined yet[8]. Since superconductivity only appears in the hidden ordered state, the mechanism of superconductivity is currently unknown[8].

The second one is the pseudogap state[9] in the copper oxide (cuprate) high $T_c$ superconductors[10]. The cuprates exhibit high $T_c$ superconductivity up to 134 K[11] when holes/electrons are doped into the CuO$_2$ plane and antiferromagnetism of the parent Mott insulator is suppressed[3]. In the low-doped regime, the pseudogap opens up at the

Fermi surface near the Brillouin zone (0, $\pi$) and ($\pi$, 0) to reduce the density of states (DOS) below a temperature $T^*$ above $T_c$[12], leaving a Fermi arc[13]. Superconductivity arises from this Fermi arc with unconventional *d*-wave spin singlet pairing[14], which is believed to have originated from Cu-3*d* based electron correlations[2–4]. The previous high-field nuclear magnetic resonance (NMR) experiment shows that the pseudogap and superconductivity coexist[12,15]. But the origin of the pseudogap, including whether any symmetry is broken below $T^*$, remains a mystery despite several decades of research[16–24].

The above-mentioned two systems are similar in that the origin of their background electronic states has not been well understood. The latter system is the focus of this paper. In the cuprates, antiferromagnetism has long been well studied in the insulating phase[2,3], but recently, long-range charge orders that break the crystal symmetry of the CuO$_2$ plane in the underdoped regime have also been revealed, whose intertwining with spin and unconventional superconductivity has attracted much attention[5].

Historically, a charge stripe order accompanied by spin order was first found in La-based cuprates with the hole concentration $p = 0.125$ below $T = 60$ K[25]. This was pinned by low-temperature tetragonal (LTT) lattice deformation and suppressed superconductivity[25]. The

[1]Department of Physics, Okayama University, Okayama, Japan. [2]Max-Planck-Institut fur Festkorperforschung, Stuttgart, Germany.
✉e-mail: kawasaki@science.okayama-u.ac.jp; zheng@psun.phys.okayama-u.ac.jp

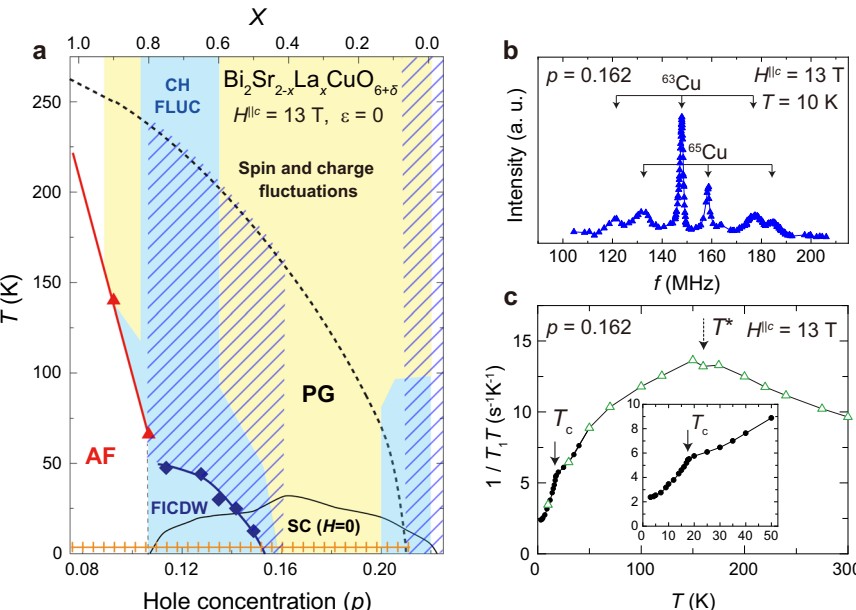

**Fig. 1 | Phase diagram of Bi$_2$Sr$_{2-x}$La$_x$CuO$_{6+\delta}$ (Bi2201) and verification of the sample. a** Doping dependence of pseudogap (PG), antiferromagnetism (AF), magnetic field-induced long-range charge-density wave order (FICDW), and superconductivity (SC) of Bi2201. Dotted and solid curves indicate pseudogap temperature $T^*$ and $T_c$. AF (spin) and CDW [charge (CH)] orders [fluctuations (FLUC)] are obtained by previous NMR measurements at $H^{\parallel c} = 13$ T[44]. The hatches indicate the regions where short-range CDW order was observed by resonant x-ray scattering[32,36,37]. The checkerboard pattern indicates the region where the checkerboard-like charge order on the surface was observed by scanning-tunneling microscopy experiments[32,40–42]. $^{63,65}$Cu-NMR spectrum at $T = 10$ K (**b**) and $^{63}$Cu-nuclear spin-lattice relaxation rate divided by temperature $^{63}(1/T_1T)$ below $T = 50$ K (solid circles) (**c**) for optimally doped Bi2201 ($p = 0.162$) are measured with cell [fixed on the cell ($V_{piezo} = 0$)]. The others (open triangles) are obtained from the bare sample [without the cell (zero strain)]. Inset of (**c**) is an enlargement of the low-temperature region below $T = 50$ K ($V_{piezo} = 0$). Dotted and solid arrows indicate $T^*$ and $T_c$, respectively.

orientation of the stripes changes by 90-degree between layers, forming a three-dimensional spin-charge stripe order in the LTT structure[25]. More recently, it has been found that magnetic field can be used as a tuning parameter of the charge order in YBa$_2$Cu$_3$O$_y$ (YBCO)[26–28] and Bi$_2$Sr$_{2-x}$La$_x$CuO$_{6+\delta}$ (Bi2201)[29]. In the latter compound, when a magnetic field above $H = 10$ T is applied to the superconducting phases over a wide doping range $p = 0.114$–$0.149$, an in-plane long-range unidirectional-incommensurate charge density wave (CDW) order appears below $T_{CDW} = 50$ - $60$ K[29]. Such magnetic field-induced long-range CDW order differs from the stripe order in that it does not involve spin order and has a different doping dependence[24]. It is also interesting that a positive correlation has been found between $T_{CDW}$ and $T^*$[29]. Resonant x-ray scattering (RXS)[30–39] and scanning-tunneling microscopy (STM) experiments[32,40–42] suggest that the magnetic field-induced long-range CDW order originated from either stripe-type ($Q_{CDW}$, 0) [(0, $Q_{CDW}$)] or checkerboard-type short-range CDW order ($Q_{CDW}$, $Q_{CDW}$), with $Q_{CDW} = 0.2$ - $0.3$ and correlation length $\xi_{CDW} = 30$ - $100$ Å at $H = 0$. In YBCO, quantum oscillation above $H = 20$ T suggests that the Fermi arcs appear to be reconstructed into small electron pockets when the long-range CDW order sets in[43].

Looking more closely into Bi2201, one notices that spin, charge, and superconductivity are intertwined. Figure 1a shows the hole concentration ($p$) vs. temperature ($T$) phase diagram of Bi2201, as determined by the previous NMR experiments at $H^{\parallel c} = 13$ T[44]. Hatched regions show short-range CDW orders observed by RXS/STM[32,36,37,40,41]. Unlike YBCO where short-range CDW order was detected by a few kHz broadening in enriched $^{17}$O-NMR spectra[45], Bi2201's Cu-NMR lacks such high resolution, hindering similar observations. Instead, charge fluctuations can be obtained from the relaxation process of the $^{63,65}$Cu nuclear spin-lattice relaxation rate ($1/T_1$)[44]; charge fluctuations coexist with spin fluctuations in the entire phase diagram[44]. The magnetic field-induced symmetry-broken long-range CDW order appears above $T_c$ in the underdoped regime $p = 0.114$–$0.149$ below the optimal

doping level $p = 0.162$. Figure 1a is reminiscent of the theoretical proposals that quantum critical fluctuations of the long-range CDW order can induce $d$-wave superconductivity[46–48].

Charge orders are often seen in the vicinity of superconductivity, not just in cuprates, but also in other strongly correlated 3$d$ electron systems, such as the layered transition metal dichalcogenides[49] and kagome vanadate CsV$_3$Sb$_5$ superconductor[50–52]. Hence, symmetry-broken charge order, together with its quantum fluctuations, are key players in the background electronic state of strongly correlated electron systems. However, a systematic understanding of their nature and their role in the occurrence of superconductivity remains elusive.

In this study, we aim to reveal the primary order that composes the background electronic state and leads to high-$T_c$ superconductivity in the pseudogap regime of the cuprates. We focused on the effectiveness of uniaxial strain for symmetry-broken electronic states. In the optimally doped Bi2201 superconductor, the long-range CDW order remains absent even after suppressing $T_c = 32$ K completely with an ultra-high magnetic field of $H^{\parallel c} = 45$ T[53]. This suggested that the long-range CDW order may only emerge near antiferromagnetism. Here we report uniaxial strain effects on the optimally doped Bi2201 superconductor ($p = 0.162$) with a tetragonal structure under a vertical field of $H^{\parallel c} = 13$ T. The sample used in this experiment and strain application are verified experimentally (see Fig. 1b, c, "Methods" and Supplementary Figs. 1–4 and Note 1–4). We find that strain symmetrically suppresses superconductivity. With increasing strain, the charge fluctuations become dominant, while the spin fluctuations are driven away. For strain above $|\varepsilon| = 0.15$ %, we find a strain-induced phase transition from short-range CDW to long-range CDW as evidenced by a peak in the nuclear spin-lattice relaxation rate divided by temperature $1/T_1T$ above $T_c(\varepsilon)$ and a pronounced increase in the linewidth of the $^{63}$Cu-NMR satellite line compared to the center line. Concomitantly, the Knight shift, reflecting the DOS at the Fermi surface, decreases upon the phase transition, indicating that the Fermi

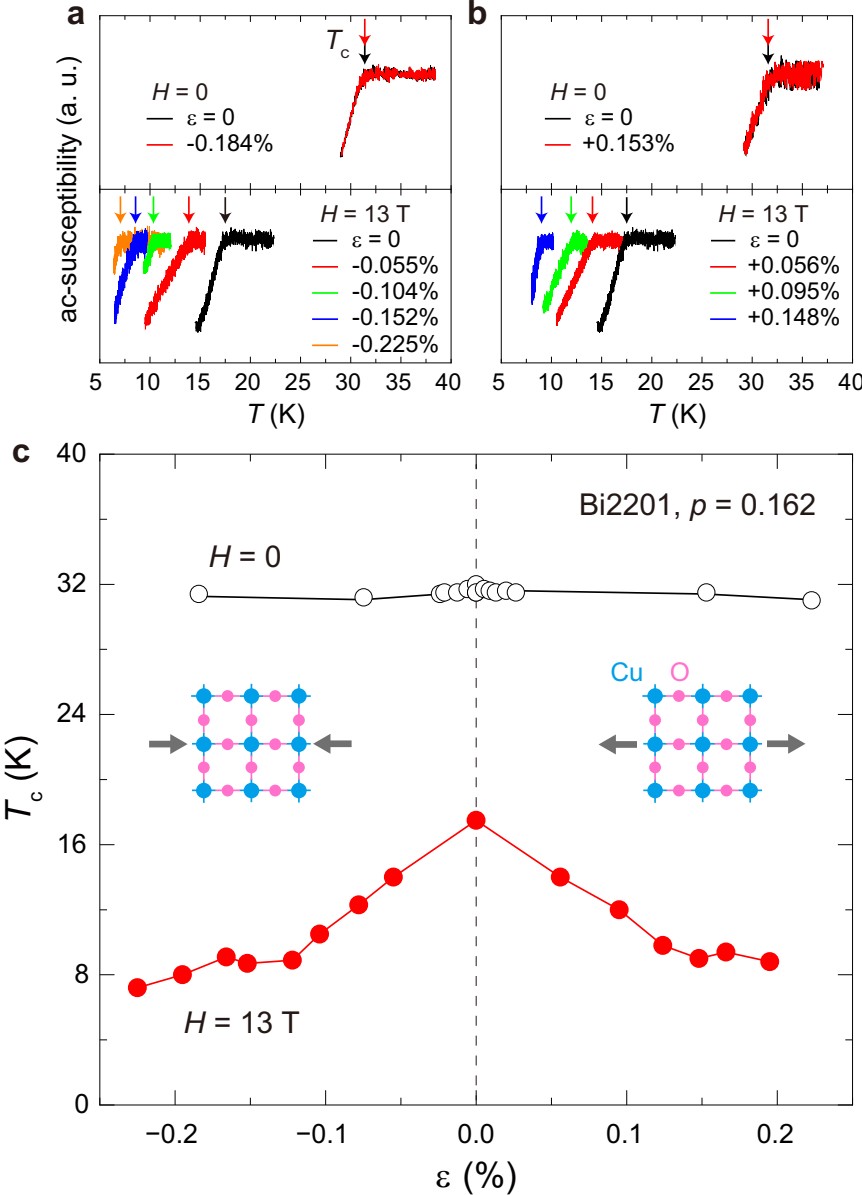

**Fig. 2 | The strain and magnetic field response of superconductivity in the optimally doped Bi2201 superconductor.** The temperature dependence of the ac susceptibility under compressive (**a**) and tensile (**b**) strains, measured at $H = 0$ and 13 T, respectively. Arrows indicate $T_c$. **c** Strain dependence of $T_c$ at $H = 0$ and 13 T. The inset shows the structure of the CuO$_2$ plane, and solid arrows indicate the applied strain direction. A dashed vertical line indicates the position of zero strain ($\varepsilon = 0$).

surface is reconstructed into a smaller one. The results suggest that a symmetry-broken long-range charge order is latent in the pseudogap regime of the cuprates as the background electronic state.

## Results

### Symmetric strain response of $T_c$

Figure 2 a, b show the temperature dependence of ac susceptibility under compressive and tensile strain measured at $H = 0$ and 13 T, respectively. The crystal broke when the strain was more than $|\varepsilon| = 0.25$ %. The strain dependence of $T_c$ is summarized in Fig. 2c. Under the composite parameters of strain and high field $H^{\parallel c} = 13$ T, $T_c$ decreases symmetrically with compression and tension, as anticipated. This observation contrasts with the highly anisotropic response observed in the underdoped YBCO superconductor with orthorhombic structure[54,55]. Because Bi2201 is a single-square CuO$_2$ plane cuprate with a tetragonal structure[56], this suggests that the present result is the

intrinsic strain response of superconductivity in cuprate. Notably, the strain response ($dT_c/d\varepsilon$) becomes small above $|\varepsilon| = 0.1$ %, and $T_c$ does not decrease appreciably. The NMR measurements were performed at $H^{\parallel c} = 13$ T below $|\varepsilon| = 0.25$ %.

### Evidence for a phase transition under strains

Figure 3 a, b show the temperature dependence of $^{63}$Cu-NMR $1/T_1T$ under various strains. First, near $T = 50$ K, $1/T_1T$ shows no clear strain response, suggesting that the pseudogap temperature $T^*$ is not strongly affected by strain. On the other hand, $1/T_1T$ decreases sharply below $T_c$ as obtained by ac susceptibility in Fig. 2.

As shown in Fig. 3a, above $T_c$, we found a peak at $T_{CDW}$ for large strains beyond $\varepsilon = -0.152$ %, while such a peak is absent for $\varepsilon = -0.104$ %. The meaning of $T_{CDW}$ will become clear later. $T_{CDW}$ increases with increasing strain, reaching $T_{CDW} = 28$ K at the maximum compression of $\varepsilon = -0.225$ %. The peak height also increases with increasing strain.

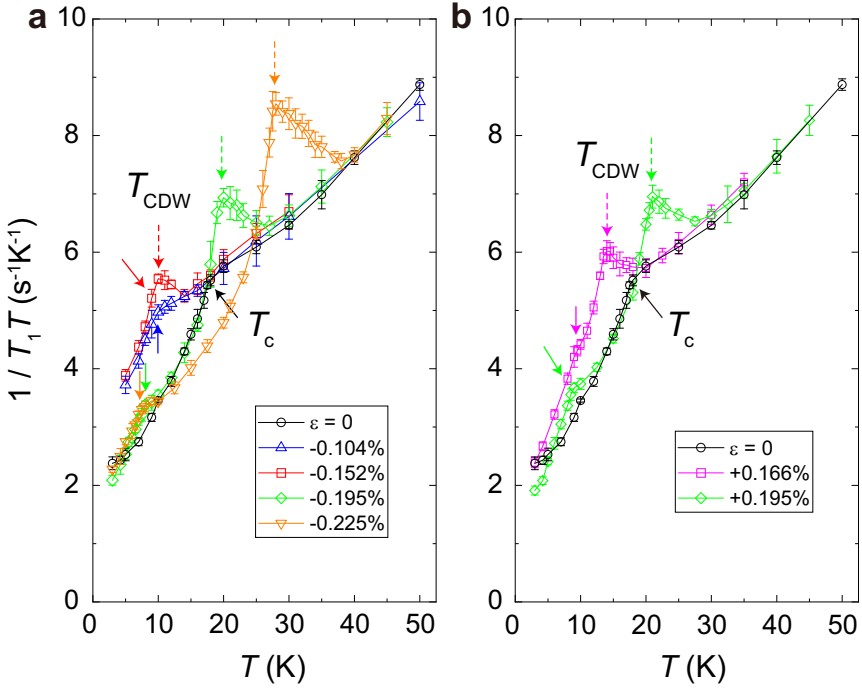

**Fig. 3 | Evidence for a phase transition in the pseudogap state under strains.** Temperature dependence of the $^{63}$Cu-nuclear spin-lattice relaxation rate divided by temperature $^{63}(1/T_1T)$ in the optimally doped Bi2201 superconductor under compressive (**a**) and tensile (**b**) strain at $H^{\parallel c} = 13$ T. Solid and dashed arrows indicate $T_c$ and $T_{CDW}$, respectively. Error bars represent the standard deviations of the fit parameters.

The situation is similar under tensile strain (Fig. 3b). In general, $1/T_1T$ shows a peak at a phase transition temperature. This occurs because the NMR relaxation rate diverges as the order parameter's fluctuation slows down towards zero at the critical point. It is noteworthy that the peak in $1/T_1T$ shows a qualitatively same behavior as the magnetic field-induced long-range CDW order found in the underdoped Bi2201 superconductors, signaling a phase transition[29,44]. We summarized $T_c$ and the observed phase transition temperature under strain in Fig. 4a.

**Strain dependence of the phase transition at $T = 10$ K**
Complementary to the insights from the temperature dependence of $T_1$, we investigated the phase transition at a fixed temperature by measuring the strain dependence of DOS and the full-width at half maximum (FWHM) of the $^{63}$Cu-NMR center line. The spin part of the Knight shift, $K_s$ (see Methods) is a direct measure of the spin susceptibility at the Cu-site and, consequently, the DOS, which correlates with the hole concentration $p^{12}$. The FWHM, on the other hand, provides information about the spatial distribution of the DOS and/or $p$.

Figure 5 displays the strain dependence of the $^{63}$Cu-NMR center line at $T = 10$ K. $K_s$ (Fig. 5b) and the FWHM (Fig. 5c) were obtained from Gaussian fitting to the spectra (see "Methods", Supplementary Fig. 4 and Note 4). $K_s$ increases with strain up to $|\varepsilon| = 0.1$ %, towards a constant, then decreases sharply beyond $|\varepsilon| = 0.15$ % (see Fig. 5b). Note that $T = 10$ K is below $T_c(\varepsilon)$ up to $|\varepsilon| = 0.1$ % (see Fig. 2c). Therefore, the increase of $K_s$ under strain reflects the recovery of the Bogoliubov quasiparticle DOS with decreasing $T_c$ up to $|\varepsilon| = 0.1$ %. The estimated relative DOS (see Methods) from $K_s$, which corresponds to the size of the Fermi arc[12,13], is also indicated with the secondary vertical axis of Fig. 5b. The reduction of the relative DOS from 0.5 to 0.3 also occurs at $|\varepsilon| = 0.15$ %, indicating that a Fermi surface reconstruction to a smaller size occurred. Here, the observation that the NMR spectrum maintains its Gaussian shape while the $K_s$ changes indicates that the strain-induced changes in the electronic state occur uniformly throughout the entire sample.

Figure 5c shows a FWHM increase of 0.2 MHz across $|\varepsilon_c| = 0.15$ %, suggesting the emergence of a distribution of the DOS, likely reflecting a spatial variation in the carrier density within the CuO$_2$ plane. Furthermore, the FWHM was found to increase above $|\varepsilon_c|$ following the strain dependence of the mean-field model, $(\varepsilon - \varepsilon_c)^{0.5}$ with $\varepsilon_c = -0.152$ %. At $T = 10$ K, the phase boundary is at $|\varepsilon_c| = 0.15$ %, as can be seen in Fig. 4a. Notably, the mean-field growth of the order parameter has also been observed in the magnetic field-induced long-range CDW order[26,29].

Therefore, the strain dependence of $K_s$ and the FWHM suggest a uniform growth of the order parameter [charge (hole) distribution amplitude] across all Cu sites above the critical strain $|\varepsilon_c|$, indicative of a long-range order (second-order phase transition) at $\varepsilon_c$.

Figure 5 d shows the data for $1/T_1T$ at $T = 10$ K as a function of strain, which were extracted from Fig. 3a, b. In contrast to the reduction of $T_c$ upon applying strain, $1/T_1T$ increases under strain. The observed behavior is qualitatively consistent with that of $K_s(\varepsilon)$. However, $1/T_1T$ continues to increase even though $T_c$ changes little and $K_s$ becomes constant above $|\varepsilon| = 0.1$ %, reaching a maximum at the phase boundary $|\varepsilon_c|$, and then decreases as $K_s$ decreases (the FWHM increases) with further strain. The peak in $1/T_1T$ at $|\varepsilon_c|$ further supports the occurrence of a long-range order (second-order phase transition).

In addition to the $T_1$ for $^{63}$Cu, we measured the $T_1$ for $^{65}$Cu at $T = 10$ K (Fig. 5d) to investigate the strain response of spin and charge fluctuations that equally exist at $\varepsilon = 0^{44}$. The strain dependence of the ratio, $^{65}(1/T_1)/^{63}(1/T_1)$, is shown in Fig. 5e. When spin fluctuations are dominant, $^{65}(1/T_1)/^{63}(1/T_1) = 1.15$, while when charge fluctuations are dominant, $^{65}(1/T_1)/^{63}(1/T_1) = 0.86$. At $\varepsilon = 0$, spin and charge fluctuations coexisted [$^{65}(1/T_1)/^{63}(1/T_1) = 1]^{44}$, however, we found that the fraction of the charge (spin) fluctuations increased (decreased) as the strain was increased and superconductivity was suppressed, and the charge fluctuations became dominant above $|\varepsilon| = 0.15$ % [$^{65}(1/T_1)/^{63}(1/T_1) = 0.86$]. Therefore, the strain-induced phase transition is most likely due to a charge-originated one. To confirm that the order parameter of

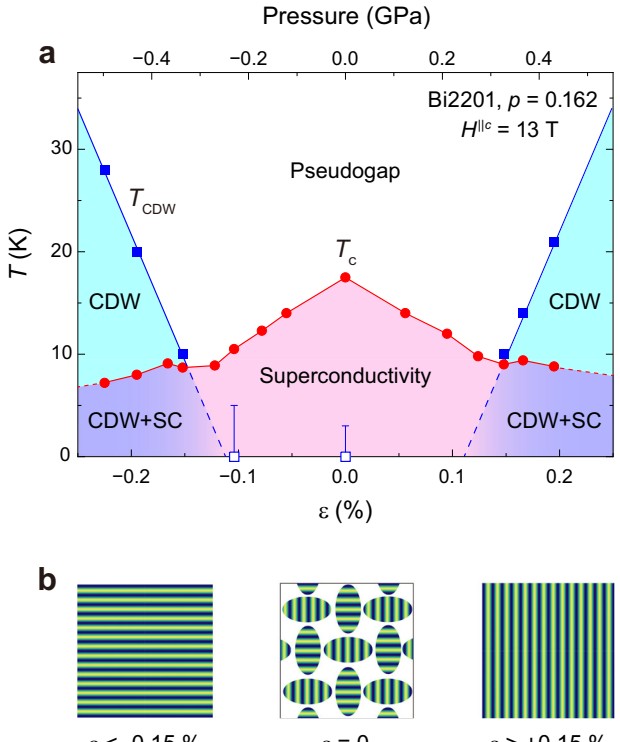

**Fig. 4 | Strain (ε) - temperature (T) phase diagram of the optimally doped Bi2201 superconductor obtained under a transverse magnetic field $H^{\parallel c}$ = 13 T. a** The strain (pressure) dependence of $T_c$ (solid circles) and $T_{CDW}$ (solid squares), respectively. CDW and SC stand for long-range charge-density wave order and superconductivity, respectively. The upper limit of the error bar indicates the lowest temperature measured. **b** The real-space image of the short-range charge-density wave domains equally in the x- and y-directions at ε = 0, observed by high-resolution resonant inelastic x-ray scattering measurement[58] and that are aligned into a long-range order in one direction (x or y) by strain above |ε| = 0.15 % (see text). The intensity of the color corresponds to the charge density.

the phase transition is indeed of charge origin, we systematically measure NMR spectra under strain.

### Evidence for a long-range charge order

Figure 6 a–f shows the temperature dependence of the $^{63}$Cu-NMR center line (Fig. 6a–c) and $^{63,65}$Cu satellite lines (Fig. 6d–f), measured under the compressive strain ε = −0.225%, zero strain ε = 0, and the tensile strain ε = +0.195 %, respectively. The signal of Cu-metal near the center peak comes from the strain cell (see, Supplementary Fig. 4).

As shown in Fig. 6g, the FWHM did not show a temperature dependence at ε = 0. However, the FWHM of the satellite line increased significantly at ε = −0.225% and +0.195%, while the FWHM of the center line increased only slightly at low temperatures. The FWHM of the satellite line increased by more than 2 MHz, compared to the center line FWHM of only 0.2 MHz, as observed in the strain dependence (Fig. 5c). In principle, the satellite line width is more affected by the distribution of the quadrupole interaction. This is because, in this experiment, the external magnetic field and the principal axis of the electric field gradient (EFG) at the Cu site are in parallel. In this situation, the center peak originated from the Zeeman interaction, while the satellite peaks are originated from both the Zeeman and the quadrupole interactions (see "Methods"). Therefore, the fact that the center peak only slightly broadens while the FWHM of the satellite peaks increases significantly below $T_{CDW}$ is evidence that a static distribution of the quadrupole interaction, rather than the Zeeman interaction, exists. Thus, Fig. 6g shows that a strain-induced long-range

CDW order occurs in the CuO$_2$ plane below $T_{CDW}(ε)$. The observed spectral changes, the satellite line being more prominently affected than the center line, are consistent with those found for magnetic field-induced long-range CDW order in underdoped cuprates[26,29,44]. Furthermore, the FWHM does not change or even increase below $T_c(ε)$, which indicates that the strain-induced long-range CDW order and superconductivity are not mutually exclusive, and superconductivity can occur even in the long-range CDW ordered state (see Supplementary Fig. 5 and Note 5 for more detail).

As can be seen in Fig. 4a which shows the obtained phase diagram of the optimally doped Bi2201 superconductor under strains at $H^{\parallel c}$ = 13 T, unlike the underdoped YBCO superconductor[54,55], we observed that both superconductivity and CDW exhibit symmetric strain responses. The present results also contrast with recent findings that stress along [100] direction has essentially no effect on $T_{CO}$ in underdoped La$_{1.875}$Ba$_{0.125}$CuO$_4$[57]. The present study demonstrates the simultaneous observation of the two orders under strains in an optimally doped cuprate.

### Discussion

First, we discuss the origin of the strain-induced long-range CDW order. In the optimally doped Bi2201 superconductor, recent high-resolution resonant inelastic x-ray scattering measurement has confirmed the formation of a short-range unidirectional stripe CDW order with equal populations of 90-degree rotated domains in the CuO$_2$ plane[58]. The wave vector and correlation length of each domain were determined, as $\mathbf{Q}_{CDW}$ = (0.23, 0) and (0, 0.23) with $\xi_{x,y}$ = 19 Å and $\xi_{y,x}$ = 10 Å, respectively[58]. Here, x and y indicate orthogonal Cu-O bond directions. NMR[45] and RXS[35,59] studies of underdoped YBCO superconductors have also suggested a unidirectional nature of the CDW order.

In this study, we applied strain along the Cu-O bond direction, which breaks the crystal symmetry of the CuO$_2$ plane. This will align the short-range CDW domains, making the wave vector parallel to one of the Cu-O bond directions. The situation is schematically illustrated in Fig. 4b. The fact that the number of domains is equal in the x- and y-directions at ε = 0[58] means that they are isotropically distributed along the Cu-O bond direction as long as the CuO$_2$ plane remains square and energetically degenerated. In other words, if strain-driven symmetry breaking lifts the degeneracy of the orthogonal domains, either domain aligns with one of the Cu-O bond directions. With increasing strain, ξ increases due to the spatial overlap of the short-range CDW order, resulting in the long-range ordering with $\mathbf{Q}_{CDW}$. Hence, $T_{CDW}$ increases with increasing strain.

We note that the strain-induced long-range CDW order is insufficient to modify the NMR satellite spectrum shape in a way, such as the peak splitting observed in the magnetic field-induced CDW order, due to the lower $T_{CDW}$[29]. But it is likely that the same incommensurate stripe-type CDW order as the magnetic field-induced one[29,44] is realized (see Fig. 4b). Assuming that the center and satellite spectra split into two peaks due to the incommensurate CDW order, similar to the magnetic field-induced long-range CDW order[29], we obtain spatial hole concentration distribution δp = ±3.5% (Supplementary Fig. 6 and Note 6), which is about half that of the magnetic field-induced CDW order. This is a reasonable value considering that the strain-induced $T_{CDW}$ is 28 K, about half of 60 K for the magnetic field-induced long-range CDW order with δp = ±6%[29]. This explanation is qualitatively consistent with the results in Fig. 5d, e, which show that the peak in 1/$T_1T$ (long-range CDW order) appears as the charge fluctuations gradually become dominant over the spin fluctuations with increasing strain. In addition, the symmetric response of the CDW under compression and tension can be understood because the domains align equally in either strain direction within the CuO$_2$ plane, due to crystal symmetry breaking. Comparison with the hydrostatic-pressure responses of superconductivity and the magnetic field-induced long-

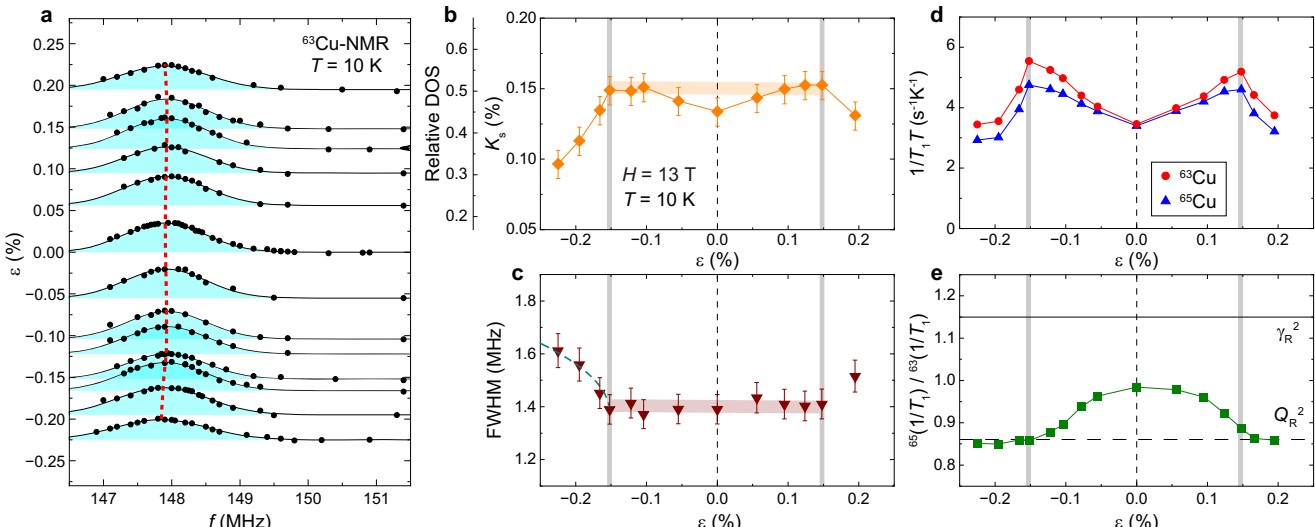

**Fig. 5 | Strain dependence of the phase transition at $T = 10$ K in the optimally doped Bi2201 superconductor. a** Strain dependence of the $^{63}$Cu center line. Shaded solid curves are the results of Gaussian fitting above 147 MHz. The vertical axis shows the strain value, and the spectrum is shifted by the strain amount from the baseline determined by fittings. The dotted curve shows the strain dependence of the peak frequency ($f_c$). **b** Strain dependence of the spin part of the Knight shift $K_s$ and relative density of states (DOS) (see Methods). **c** Strain dependence of the full-width at half maximum (FWHM). The dashed curve is the fitting result of a mean-field model, with FWHM = $1.380 + 0.832 \times (\varepsilon - \varepsilon_c)^{0.5}$ and $\varepsilon_c = -0.152$ %. The

shaded horizontal lines serve as visual guides. Error bars represent the standard deviations of the fit parameters. **d** Strain dependence of the nuclear spin-lattice relaxation rate divided by temperature $1/T_1T$ at $T = 10$ K for $^{63}$Cu (circles) and $^{65}$Cu (triangles), respectively. **e** Strain dependence of the ratio, $^{65}(1/T_1)/^{63}(1/T_1)$. The solid and dashed horizontal straight lines correspond to the values expected for the magnetic relaxation process [$\gamma_R^2$ (= 1.15)] and for the quadrupole relaxation process [$Q_R^2$ (= 0.86)], respectively. Dashed vertical lines indicate $\varepsilon = 0$. Solid vertical lines indicate the phase boundary $|\varepsilon_c|$ under strain at $T = 10$ K. The error in $T_1$ is on the order of the symbol size.

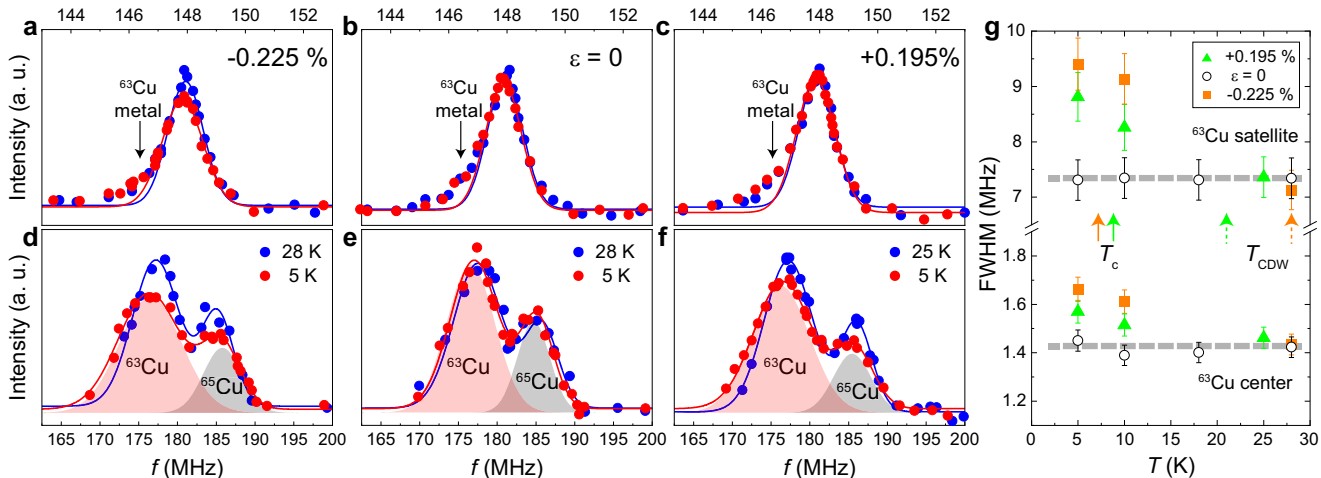

**Fig. 6 | Evidence for a strain-induced long-range charge order at $T_{CDW}$.** Temperature dependence of Cu-NMR spectra of the optimally doped Bi2201 superconductor under strain at $\varepsilon = -0.225$ % (**a**, **d**), $\varepsilon = 0$ (**b**, **e**), and $\varepsilon = +0.195$ % (**c**, **f**). **a**–**c** show the results of the $^{63}$Cu center and **d**–**f** show those of the $^{63,65}$Cu satellite lines. The solid line is the result of fitting with a Gaussian function. For the center line, Cu metal is excluded to obtain the full width at half maximum (FWHM). Two

Gaussian fits are employed to obtain the FWHM of the $^{63}$Cu satellite line. The solid curve is the sum of the two Gaussian functions for the $^{63}$Cu and $^{65}$Cu satellites (shaded area). **g** Temperature dependence of the FWHM of the $^{63}$Cu center and satellite lines. The solid and dashed arrows indicate $T_c(\varepsilon)$ and $T_{CDW}(\varepsilon)$, respectively. At $\varepsilon = 0$, $T_c = 17.5$ K. The dashed horizontal lines serve as visual guides. Error bars represent the standard deviations of the fit parameters.

range CDW order supports this hypothesis. In the underdoped YBCO superconductor, although $T_c$ slightly increases in the opposite trend to the strain response in the plane, the magnetic field-induced long-range CDW order is suppressed under high hydrostatic pressure up to $P = 2$ GPa, which preserves crystal symmetry[60].

It is emphasized that the long-range CDW order is absent in the optimally doped Bi2201 superconductor without strain even under an ultra-high magnetic field of $H^{\parallel c} = 45$ T[53], while it appears in the underdoped YBCO superconductor at high fields[26] or under strain at $H = 0$[54]. Therefore, it can be concluded that a crystal symmetry breaking in the

CuO$_2$ plane is the most important factor for the short-range CDW to develop into a long-range order, as observed in this study.

Next, we discuss the relationship between superconductivity and the long-range CDW order. As summarized in Fig. 4a, $T_c$ is reduced by strain at $H^{\parallel c} = 13$ T. In addition, the doping (Fig. 1a) and strain (Fig. 4a) phase diagram suggest that $T_c$ decreases when the charge fluctuations replace the spin fluctuations regardless of doping or strain, suggesting that the charge fluctuations tend to be unfavorable for superconductivity. In fact, recent theories also suggest that CDW is a competitor of superconductivity[61]. Upon the emergence of long-range CDW

order above $|\varepsilon| = 0.15$ %, where a reduction in the DOS suggests Fermi surface reconstruction (Fig. 5b), $T_c$ is reduced but retains a finite value.

The nature of superconductivity in the long-range CDW ordered state has been debated[5]. As depicted in Fig. 3a, b, the temperature dependence of $1/T_1T$ below $T_c$ remains largely unchanged under strain, suggesting that the superconductivity may retain its $d$-wave nature in the long-range CDW ordered state. Therefore, it is suggested that the reconstructed Fermi surface is compatible with $d$-wave superconductivity ($d$-SC), or the superconducting order parameter adapts to the spatial modulation of the charge density, which is why $dT_c/d\varepsilon$ becomes small after the long-range CDW order appears. Below, we discuss two adapted forms of superconductivity. In the first place, $d$-SC can coexist with unidirectional long-range CDW order as a form of a striped superconducting state[5]. Second, in the underdoped region without strain, a $p \cdot T$ phase diagram has been proposed in which superconductivity exhibits as a form of a pair density wave (PDW) within a long-range CDW ordered state[62,63]. A PDW state is a superconducting state with a spatially modulated electron-pair density and energy gap with an average value of zero[5,64]. For example, a unidirectional PDW ordered state has a wave vector half that of the preexisting long-range CDW ordered state, i.e., $Q_{PDW} = Q_{CDW}/2$[5].

A central challenge in this regard is to experimentally distinguish between the ordinary long-range CDW order + $d$-SC coexisting state and the long-range PDW state, as these states are energetically nearly degenerate[5]. NMR could potentially capture the difference in spatial modulation between these two states below $T_c$. However, no change in the spectral shape indicative of a PDW state was observed below $T_c$ (see Fig. 6 and Supplementary Fig. 5). This is consistent with previous NMR studies on YBCO that also did not find conclusive evidence for a PDW state[65].

Notably, a possible PDW state is currently being widely discussed in strongly correlated electron superconductors[64], including cuprate superconductor $Bi_2Sr_2CaCu_2O_{8+x}$[66], kagome superconductor $CsV_3Sb_5$[67], heavy fermion superconductor $UTe_2$[68], and iron-based superconductor $EuRbFe_4As_4$[69]. However, experimental observations have thus far been confined to surface-sensitive STM techniques. Therefore, to demonstrate a PDW state in bulk, a different approach is required. NMR experiments, such as the one presented here, are a promising candidates. We anticipate that the strain experiment in this investigation will pave the way for future research into the PDW state.

In conclusion, through the measurements of ac susceptibility, $^{63,65}$Cu-NMR $T_1$, and spectra in the optimally doped Bi2201 superconductor under uniaxial strain, we found that strain-driven crystal symmetry breaking enhances CDW and suppresses superconductivity in the pseudogap state even in the optimally doped regime that is apart from the antiferromagnetically-ordered region or the magnetic field-induced long-range CDW ordered region. Under the strain $|\varepsilon| = 0.15$% along the Cu-O bond direction, charge fluctuations become dominant over spin fluctuations followed by a suppression of superconductivity. For higher strain above $|\varepsilon| = 0.15$%, we found a strain-induced long-range CDW order with a reduction of the DOS likely due to Fermi surface shrinking. The present results suggest that a long-range CDW order is a latent order in the pseudogap phase, which becomes visible by a slight crystal symmetry breaking. Superconductivity survives in the long-range CDW ordered state with possible reconstructed Fermi surface, suggesting the possibility of exotic nature such as a pair density wave state. The present results highlight the importance of the CuO$_2$ plane symmetry in the cuprates and suggest that uniaxial strain could be a valuable tool to further elucidate the intertwined orders in strongly correlated electron superconductors.

## Methods
### Sample
$Bi_2Sr_{2-x}La_xCuO_{6+\delta}$ crystallizes tetragonal structure (space group $I4/mmm$) without a structural phase transition[56]. There is only one CuO$_2$ plane per unit cell, and the interplanar spacing is the largest in cuprates. The single crystal of $Bi_2Sr_{1.6}La_{0.4}CuO_{6+\delta}$ ($p = 0.162$, $T_c = 32$ K) was grown by the traveling solvent floating zone method[56]. The hole concentration ($p$) was estimated previously[70]. The single crystal plates used for the measurement were cut from the same ingot as the crystal used in the previous NMR experiments[12,15,29,53]. Small and thin single-crystal platelet, sized up to $L_0 \times W \times th = 2.00 \times 0.62 \times 0.12$ mm$^3$ where $L_0$ is the length of the strained part, $W$ is the width, and $th$ is the thickness along the $c$ axis was cleaved from the ingot. The strain was applied along the Cu-O bond direction, which was determined by the Laue picture. The $^{63,65}$Cu-NMR spectrum (Fig. 1b) and $^{63}(1/T_1T)$ below $T = 50$ K (Fig. 1c) measured at $V_{piezo} = 0$ for reproducibility are consistent with the results measured on the bare sample. $T^* \approx 160$ K and $T_c(H^{\|c} = 13$ T$) = 17.5$ K are consistent with the previous results[12,15,29,44,53].

### NMR
There is only one Cu site in the single CuO$_2$ layer cuprate Bi2201. For $^{63,65}$Cu-NMR, the Hamiltonian is the sum of the Zeeman and nuclear quadrupole interactions[71] as $\mathcal{H} = \mathcal{H}_z + \mathcal{H}_Q = -^{63,65}\gamma\hbar\mathbf{I} \cdot \mathbf{H}_0(1 + ^{63,65}K) + \frac{h^{63,65}\nu_Q}{6}[3I_z^2 - I(I+1) + \eta(I_x^2 + I_y^2)]$, where gyromagnetic ratio $^{63}\gamma$ ($^{65}\gamma$) = 11.285 (12.089) MHz/T, $^{63,65}K$ is the Knight shift, and the nuclear spin $I = 3/2$. $^{63,65}\nu_Q = \frac{3eqV_{zz}}{2I(2I-1)h}\sqrt{1 + \eta^2/3}$ with $^{63}Q$ ($^{65}Q$) = $-0.220(-0.204) \times 10^{-24}$ cm$^2$ [72] and $V_{zz}$ being the nuclear quadrupole moment and the EFG tensor[71]. The principal axis of the EFG is along the $c$ axis, which is parallel to the external magnetic field $\mathbf{H}_0$. The asymmetry parameter $\eta$ are defined as $\eta = (|V_{yy}| - |V_{xx}|)/|V_{zz}|$, which is zero[73] and the strain (up to 0.225 %) in this study was not large enough $\eta$ to be clearly appeared in the measurement results. The $^{63,65}$Cu-NMR spectra were taken by sweeping the rf frequency by using a phase-coherent spectrometer. As shown in Fig. 1b, when $\mathbf{H}_0$ and the principal axis of the EFG are in parallel, the center and the two satellite lines between $|m\rangle$ and $|m-1\rangle$, ($m = 3/2, 1/2, -1/2$), at the resonance frequency $f_{m\leftrightarrow m-1} = ^{63,65}\gamma H_0(1 + ^{63,65}K) - (^{63,65}\nu_Q)(m - 1/2)$ are obtained for $^{63}$Cu and $^{65}$Cu, respectively[71].

$^{63}K$(%) is obtained from the peak frequency ($f_c$) of the center line as $K(\%) = 100 \times \frac{f_c - \gamma H_0}{\gamma H_0}$. The Knight shift is denoted by $K = K_s + K_{orb}$, where $K_s$ and $K_{orb}$ are the spin and orbital contributions, respectively. $K_s$ is proportional to the spin susceptibility $\chi_s$[71] and thus reflect the DOS. $K_{orb}$ is known to be 1.21 %[12]. Therefore, we can estimate the relative DOS at $T = 10$ K as $\frac{N(10K)}{N_0} = \frac{K_s(10K)}{K_s(T \geq T^*)} = \frac{[K(\varepsilon) - 1.21\%]}{0.3\%}$, where $N_0$ is the DOS above $T^*$ and $K_s(T \geq T^*) = 0.3$ % was previously determined[12].

In the long-range CDW ordered state, a spatial modulation $\delta\nu_Q(\mathbf{r})$ [$\gg \delta K(\mathbf{r})$] appear to split or broaden the spectrum[29,74,75]. The $^{63}$Cu-NMR high frequency satellite line ($-1/2 \leftrightarrow -3/2$ transition, $f \approx 177$ MHz) is used to detect a long-range CDW order in the same way as in the previous report of the magnetic field-induced long-range CDW order[29].

The $^{63,65}$Cu-NMR $T_1$ were measured at the center peak ($+1/2 \leftrightarrow -1/2$ transition, $f \approx 148$ and 158 MHz, respectively). To obtain $T_1$, the time dependence of the spin-echo intensity after the saturation of the nuclear magnetization $M$ (recovery curve) was fitted by the theoretical function[76]. Typical recovery curves used to determine $T_1$ are shown in Supplementary Fig. 7 and Note 7. In general, in strongly correlated electron systems, $T_1$ probes the spin fluctuation through the hyperfine coupling constant $A_\mathbf{q}$ as $^{63,65}(1/T_1^M) \propto ^{63,65}\gamma k_B T \sum_\mathbf{q} |A_\mathbf{q}|^2 \chi''_\perp(\mathbf{q},\omega_0)/\omega_0$, where $\omega_0$ is the NMR frequency and $\mathbf{q}$ is a wave vector for a spin order[77]. On the other hand, in the case that $T_1$ probes the EFG (charge) fluctuation, then it is dominated by the quadrupole moment as $^{63,65}(1/T_1^Q) \propto 3(2I+3)(^{63,65}Q^2)/[10(2I-1)I^2]$[78]. Therefore, the origin of the nuclear spin lattice relaxation process for the cuprate can be identified from the ratio, $^{65}(1/T_1)/^{63}(1/T_1)$[44]. When spin fluctuations dominate, $^{65}(1/T_1)/^{63}(1/T_1) = (^{65}\gamma/^{63}\gamma)^2 = \gamma_R^2 = 1.15$, while charge fluctuations dominate, $^{65}(1/T_1)/^{63}(1/T_1) = (^{65}Q/^{63}Q)^2 = Q_R^2 = 0.86$[44]. In the

superconducting state below $T_c$, $1/T_1T$ reflects the temperature dependence of the quasiparticle density of states, $N(E_F)$. The $T_1$ measurements under strain in this study were performed in the pseudogap state below $T = 50$ K. If a long-range CDW order occurs, $1/T_1T$ shows a peak at $T_{CDW}$[29].

## Uniaxial strain

A homemade piezoelectric-driven strain cell is used to compress and/or tensile single crystal plate. Following C. W. Hicks et al.[79], as shown in Supplementary Fig. 1, we have arranged three commercially available piezoelectric actuators (PI, P-885.51) in a configuration of two outer and one inner actuators, and connected them with a titanium cell to allow compression and tension of the crystal plate in the center of the cell. A sample plate was fixed to the cell using epoxy (STYCAST 2850FTJ). A power supply (Razorbill Instruments, RP100) was used to drive the actuators. The actual displacement is determined by reading the capacitance of a homemade parallel-plate capacitor placed next to the sample with an LCR meter (Keysight, E4980AL). For NMR measurement, to avoid temperature-dependent changes in strain, we applied voltage to the actuators and monitored the strain value at each temperature.

In this paper, in line with previous strain experiments[57,80,81], we define strain ($\varepsilon$) using the displacement $x$ as $\varepsilon(\%) = 100 \times (x - x_0) / L_0$, where $L_0$ is the unstrained length of the crystal. We have verified that $x_0$ ($V_{piezo} = 0$) corresponds to the zero-strain (bare sample) within the error of NMR experiments and $T_c$ measurements, based on the results of our investigation using Bi2201 (see Fig. 1c and Supplementary Fig. 1-4 and Note 1-4). The results measured with the sample fixed with the cell at zero volts ($V_{piezo} = 0$) all agree with the results measured with the bare sample (zero strain). As shown in Supplementary Fig. 2, Hooke's law was confirmed within the strain range of the experiment, indicating that the strain was uniformly distributed in the sample.

Our cell does not have a force sensor to directly measure pressure alongside strain. Therefore, for reference, we estimated pressure values using the elastic constants of $Bi_2Sr_2CuO_6$ from prior first-principles calculations[82]. These estimated values are shown on the upper axis of Fig. 4a. Of note, our value is in good agreement with that reported in strain experiments on the related cuprate $La_{1.875}Ba_{0.125}CuO_4$ superconductor[57].

## Measurements

We measured ac susceptibility, $^{63,65}$Cu-NMR spectrum, and nuclear spin-lattice relaxation time $T_1$. A high-resolution superconducting magnet (Oxford Instruments, AS600) is used to apply a constant magnetic field of $H = 12.951$ T ($H^{\parallel c} = 13$ T in this paper) along the $c$ direction to partially suppress superconductivity and to compare the results with those at $\varepsilon = 0$ (Fig. 1a). The high field is also effective for NMR experiments on a tiny single crystal, as it can improve the signal-to-noise ratio. $T_c(\varepsilon)$ was measured by recording the resonance frequency of the NMR coil during both cooling and warming the sample on the strain cell. To detect a long-range charge order, $^{63}$Cu-NMR $T_1$, the center, and the satellite spectra were measured systematically. The strain dependence of $T_c$ was obtained by using six single-crystal plates and three homemade strain cells (#3, #4, and #5). This was done to confirm the reproducibility of the results and to determine the strain amount at which Hooke's law holds. The results for all six sample plates were the same until the plate broke, so the results are shown without distinction in Fig. 2. NMR experiments were performed on the sixth plate. The photographs in Supplementary Fig. 1 show cell #5.

## Data availability

All data supporting the findings of this work are presented in the manuscript and its associated Supplementary Information. Additional data are available from the corresponding author(s) upon request.

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

## Acknowledgements

We thank S. Yoshida for his effort in developing the homemade strain cell during the early stage of this work. This work was supported by JSPS KAKENHI Grant Numbers JP19H00657 (G.-Q.Z.), JP19K03747 (S.K.), and JP23K03323 (S.K.) and a research grant from the Murata Science and Education Foundation (S.K.).

## Author contributions

G.-Q.Z. supervised the project. C.-T.L. synthesized and characterized the single crystal. S.K. and N.T. performed susceptibility and NMR measurements under strain. S.K. and N.T. analyzed and visualized the data. S.K. and G.-Q.Z. wrote the paper. All authors discussed the results and interpretation.

## Competing interests

The authors declare no competing interests.
