## [Peer Review File · Nature Communications]

Strain-induced long-range charge-density wave order in the optimally doped $\text{Bi}_2\text{Sr}_{2-x}\text{La}_x\text{CuO}_6$ superconductorREVIEWER COMMENTS

Reviewer #1 (Remarks to the Author):

The paper describes the discovery that uniaxial strain greatly strengthens CDW order in the cuprate Bi2201, at optimal hole doping for superconductivity.

This is a beautiful set of data and an important result which, in principle, warrants publication in a high-profile journal such as Nature Communications. However, the current version of the manuscript contains significant flaws that must be rectified before I can recommend publication. Primarily, the manuscript crucially lacks precision and the authors tend to extrapolate well beyond what is reasonable to deduce from the data. This is both unnecessary and actually detrimental to the overall credibility of the paper.

1) First of all, there is a problem with the title and with all the sentences claiming that CDW order is induced by strain or that "CDW is absent in the optimally doped Bi2201" (page 9, for instance). In fact, short-range CDW order is observed without strain by both X-ray diffraction (ref 49) and STM (please cite works, at least the PRX paper from Hoffman's group). The authors do not detect it with NMR, presumably because of broad NMR lines. Strain strengthens but does not induce the CDW.

2) Similarly, the authors state that what they see is long-range order. I agree that this is what likely happens. However, they cannot prove it because NMR is a local technique and furthermore, they only see a broadening, not any particular shape of the NMR line. Therefore, they cannot exclude that strain increases primarily the amplitude of the CDW. The authors should be more precise and more careful on this point, especially as they see a splitting at lower doping. Thus, I am tempted to conclude that the CDW under strain is not as long ranged as the CDW at lower doping in high fields.

3) The following sentence is problematic in several respects, especially within the abstract: "the findings reveal that CDW is an underlying hidden order in the pseudogap state, not limited to the underdoped regime where it is magnetically induced, becoming apparent upon symmetry breaking of the CuO₂ plane by strain".

- CDW is no longer hidden since it is seen by many probes, including in the studied compound by x-rays and STM. So, from this standpoint the present study "reveals" nothing.

- Also, STM studies in Bi-based cuprates already reported the CDW to persist in overdoped samples, actually throughout the pseudogap regime.

- As far as I know, the origin of the cuprate CDW is still a controversial issue so I am wondering what leads the authors to assert that it is "magnetically induced" in the underdoped regime.

- To me, the intriguing and interesting outcome of this work is the effectiveness of a slight orthorhombic distortion to stabilize strong CDW order. This is what the authors should focus on.

4) Abstract:

- "this sheds light on the physics of cuprates": rather than such a vague sentence, one would like to know precisely what the results tell us on the physics of cuprates.

- "and may have implications for other strongly correlated electron systems": this kind of general, unprecise statement should be avoided in my opinion.

- Also, on page 11, the authors write "The present results (...) demonstrates that uniaxial strain is a promising tool to elucidate the intertwined orders in strongly correlated electron superconductors". The verb "demonstrate" may sound slightly over-emphatic. Uniaxial strain has become a well-established technique in the field of correlated electron systems, with numerous published studies showcasing its effectiveness and yielding remarkable results. In fact, the present results show that the technique has the potential to contribute better understanding of the cuprates, which is already a most remarkable achievement.

5) The whole story about the DOS is unclear and unconvincing:

- it requires to dive into Ref. 12 to understand the normalization factor 0.5 and the notion of relative DOS (that is not defined in the text). Actually, I don't find this relative DOS to be a useful notion here.

- it is far from being proven, and even quite doubtful, that T1 values reflect the DOS. Due to antiferromagnetic fluctuations, the relaxation rate does not follow the Korringa law of Fermi liquids and furthermore there is quadrupole contribution to T1.
- I am not sure how to make sense of a comparison of T1 values at Tc(epsilon), i.e. at different temperatures.
- I find it very difficult to be convinced that this data provides any evidence of Fermi surface reconstruction.
- The authors should rather consider as a reasonable assumption that the Fermi surface gets reconstructed as strain strengthens the CDW (although the issue as to how the CDW correlation length affects the Fermi surface reconstruction is a complicated one).

6) Some effort in sharpening the content is necessary. There are many repetitions and many things that could be written in a much more compact way. This would greatly improve readability.

- On page 7 notably, there are many sentences to say that strain induces stronger CDW order while depressing but not suppressing superconductivity: "a long-range CDW order orders in the CuO2 plane", "the order parameter (...) is of charge origin", "induces a CDW order".
- On the same page, the sentence "the bottom panel in Fig. 6a (...)" is useless here as it is discussed in another section.
- Page 10 "a PDW state is a superconducting state with a spatially modulated electron-pair density and energy gap" and "the superconducting order parameter in a PDW state exhibits spatial variations (...)".
- Etc.

7) Page 7. "such changes in the spectrum are perfectly consistent with those observed in (...)". The authors should be more precise. They refer to studies where a line splitting is seen, while they observe a broadening instead of a splitting here. So, it is not immediately obvious what is consistent with what.

8) Page 8. In YBCO, an xray scattering study without strain has already found that the short-range CDW consists of domains of unidirectional CDW: Comin et al. Science 347, 1335 (2015). Also, an NMR study reported that the short-range CDW shows in-plane anisotropy at the local scale: Wu et al. Nature Communications 6, 6438 (2015).

9) "under strain, despite the lack of evidence of long-range order". This should be slightly reformulated so that readers don't get the impression that strain fails to induce long-range order in YBCO, which is obviously not true. I think it is sufficient to mention here that studies of the short range CDW in YBCO have also concluded that it is locally unidirectional (above refs. + ref. 50.).

10) Page 8. "lifts degeneracy of energy levels". A more specific expression than "energy levels" would probably be preferable. Degeneracy of the orthogonal domains?

11) Page 9. Hydrostatic pressure: the transport paper cited by the authors (ref 51) appears to be in contradiction with Xray scattering data: Souliou et al. PRB 97, 020503(R) (2018). There is actually an NMR study of Vinograd et al. about the hydrostatic pressure effect on the CDW in YBCO that does a great job summarizing this issue with all the relevant references: Phys. Rev. B 100, 094502 (2019).

12) Fig. 6b is quite difficult to read. A rotation of the (p,T) plane would probably help. However, I am afraid there is too much information to make things really clear here. Are the large blue areas depicting CDW fluctuations necessary? To me this is rather confusing. Furthermore, I am afraid this is misleading as these blue areas depict situations in which quadrupole relaxation is particularly strong. However, my understanding of ref. 37 is that quadrupole relaxation, i.e. CDW fluctuations, are present at essentially all other dopings as well. Also confusing is that the phase diagram ignores STM and Xray works reporting the presence of short-range CDW order throughout the pseudogap regime, including the optimal doping discussed here. These remarks obviously apply to Fig. 1a as well.

13) Page 9: "This demonstrates that the CDW order is accompanied by the Fermi surface reconstruction in cuprates". I don't get what demonstrates that CDW order is accompanied by

Fermi surface reconstruction. Furthermore, the sentence sounds like a general conclusion applying to all cuprates while the authors can only make statements about Bi2201. Actually, why would the Fermi surface not be reconstructed? If there is well defined CDW order, it has to be reconstructed, it is not really an issue.

14) The authors recognize they have nothing to say about PDW order from the present data. Yet, there is more than one page about PDW. I find this unreasonable. Concerning the prospect of addressing this question with NMR, the authors might notice that this has been done by Vinograd et al. Nat. Commun. 12, 3274 (2021) in the context of YBCO.

15) Page 3: "Resonant x-ray scattering (RXS) experiments suggest that the field-induced CDW order found by NMR (...)". The sentence could suggest that the high-field CDW has been seen only by NMR, which is incorrect: it has been also seen by Xray scattering in YBCO (Science 350, 949–952 (2015) 914–915 and 3 or 4 subsequent papers).

16) Page 3: "which suggested the necessity of completely suppressing superconductivity for the charge-ordered state to emerge in this compound". This is incorrect. Superconductivity and CDW coexist over some field range in YBCO. See Wu et al. Nat Commun 4, 2113 (2013) as well as the phase diagrams in these two papers: Kacmarcik et al. Phys. Rev. Lett. 121, 167002 (2018) and LeBoeuf et al. Nat. Phys. 9, 79–83 (2013). It is also true in the La214 cuprates.

17) Page 9: "The present results suggest that superconductivity can survive even though the Fermi surface is reconstructed and/or the static CDW order is present". Same remark as preceding points concerning the SC – CDW coexistence and about the Fermi surface reconstruction.

18) Page 4: CDW in the infinite layer nickelates is apparently quite controversial to say the least. People now think the signal is due to oxygen ordering. There are probably better examples to find.

19) In Fig. 5, I initially wondered why the signal from Cu metal is that broad. After reading the Methods, I understood the signal must come from the strain cell as the authors used silver coils. Is this correct? Better specifying the origin of the Cu-metal signal in the caption.

20) Methods section: "Three homemade strain cells (#3, #4, and #5) were used in parallel to perform all the strain experiments." I am not entirely sure what is the purpose of this information and what "in parallel" precisely means here.

Reviewer #2 (Remarks to the Author):

This manuscript describes Cu NMR studies of a prototypical high T_c cuprate under uniaxial strain. This material is particularly nice because it has a tetragonal crystal structure, unlike other cuprates that are orthorhombic. By applying strain, the C₄ symmetry of the lattice is broken and the authors find that charge ordering is stabilized, while superconductivity is slightly suppressed. The results are interesting, but there are a number of issues with the manuscript that need to be addressed.

(1) The strain values reported do not take into account thermal contraction. The authors report the strain based on length, L, as measured by capacitance, and the original length, L₀. The latter, however, will be temperature dependent because of the thermal contraction of the crystal. When the temperature is reduced the crystal will likely be under finite tensile strain even in the absence of any applied force, and that "epsilon = 0" in Figs. 2, 3, 4, and 6, is not really zero strain. Other publications have used an intrinsic measurement that is sensitive to strain in order to determine the zero strain value of L, or used measurements of the applied stress to infer the strain. These include <https://doi.org/10.1103/PhysRevB.108.205113>, <https://www.nature.com/articles/s41467-018-03377-8>, and <https://www.nature.com/articles/s41586-019-1596-2>. None of these other works are cited by the authors, however, and there is no discussion of how to deal with the issue of determining epsilon = 0 at cryogenic temperatures.

(2) The manuscript contains several incorrect, vague, or misleading statements. For example, the abstract states that strain "deliberately breaks translational and rotational symmetries". In fact, strain breaks a discrete rotational symmetry but translation symmetry is already broken in a crystalline lattice even without strain. In the introduction, the authors state that URu₂Si₂ and the high T_c cuprates are "examples of background electronic states with broken symmetry", referring to the hidden order and 'pseudogap' states. However, they also state that it is "unknown whether any symmetry is broken" in the latter. These statements are contradictory and confusing. Furthermore, they state that long-range charge order breaks translation symmetry. Presumably they mean that it breaks discrete translation symmetry of the crystalline lattice, since any solid lattice already breaks translation symmetry. It is also unclear why they state that the triple Q charge order in CsV₃Sb₅ breaks time reversal symmetry. In the discussion, they state that "high magnetic field and uniaxial strain are effective in breaking new ground in physics of strongly correlated electron systems". This statement seems vague and unnecessary.

(3) There is no discussion of local strain by dopant atoms. They state that strain is applied along the Cu-O bond direction, "which breaks the local crystal symmetry of the CuO₂ plane". Since the strain field is presumably long range, it is not clear why the symmetry breaking is local. They also refer to "artificial local symmetry breaking". Local symmetry breaking can be caused by the dopants themselves, leading to a random strain field. This leads to an important point: how large is the local strain field from the dopants, versus the externally applied strain field? The authors point out that they see no change in the EFG asymmetry parameter under strain. But couldn't the changes they observe in the quadrupolar satellites reflect such a change? In other words, what would happen to the EFG parameters in the presence of the charge order that the authors hypothesize exists? Wouldn't it naturally give rise to a non-zero η ? Presumably the spectra are broad because of a distribution of local strains and EFGs, and the external strain field only slightly modifies this distribution.

(4) They point out that the CDW order in the Bi2201 sample is insufficient to lead to peak splitting in their NMR spectra "due to the lower T_{CDW}". Presumably they are assuming that the magnitude of the CDW order parameter is proportional to T_{CDW} when they make such a statement, but they should clarify this point.

(5) The authors need to address their results in the context of other NMR studies of cuprates under strain. How does one reconcile the fact that strain seems to induce static charge order in Bi2201, but has essentially no effect on charge order in the 214 system (see 10.1103/PhysRevB.108.205113)?

RESPONSES to Reviewers' remarks

We would like to express our sincere gratitude to both Reviewers for their time, thorough and insightful comments, and constructive suggestions. Their feedback has greatly improved the quality of our manuscript.

We have carefully considered each comment and have made the following revisions to the manuscript:

Responses to the remarks by Reviewer #1:

The paper describes the discovery that uniaxial strain greatly strengthens CDW order in the cuprate Bi2201, at optimal hole doping for superconductivity. This is a beautiful set of data and an important result which, in principle, warrants publication in a high-profile journal such as Nature Communications.

We are grateful to the Reviewer for his/her high appreciation of the quality of our experimental data and the importance of our results.

However, the current version of the manuscript contains significant flaws that must be rectified before I can recommend publication. Primarily, the manuscript crucially lacks precision and the authors tend to extrapolate well beyond what is reasonable to deduce from the data. This is both unnecessary and actually detrimental to the overall credibility of the paper.

We appreciate the Reviewer's suggestions of revision. In the initially submitted version, there were indeed ambiguous expressions regarding the CDW order that could mislead readers. We have revised the manuscript to use clearer and more precise expressions. We have also added new experimental results (revised Fig. 5a-c) that directly support our conclusions, replacing the previous Fig. 4c.

Below, we address each of the Reviewer's comments point by point.

1) First of all, there is a problem with the title and with all the sentences claiming that CDW order is induced by strain or that "CDW is absent in the optimally doped Bi2201" (page 9, for instance). In fact, short-range CDW order is observed without strain by both Xray diffraction (ref 49) and STM (please cite works, at least the PRX paper from Hoffman's group). The authors do not detect it with NMR, presumably because of broad NMR lines. Strain strengthens but does not induce the CDW.

The Reviewer is correct that the initial manuscript contained un-precise statements regarding CDW. We do not deny the existence of short-range CDW order observed by XRD. Our intended conclusion is that we discovered that short-range CDW order undergoes a phase transition to long-range CDW order under strain. While a long-range CDW order remains unseen in optimally doping level of any cuprates, our study demonstrates for the first time that strain can induce its emergence in Bi2201.

In the revised manuscript, we have clearly distinguished between the terms "short-range CDW order" and "long-range CDW order," including the fact that short-range CDW order in Bi2201 cannot be observed with our NMR resolution, so that the situation is clear to the reader. To concisely and clearly state the conclusion of our paper, we have also added the word "long-range" to the title of the paper.

2) Similarly, the authors state that what they see is long-range order. I agree that this is what likely happens. However, they cannot prove it because NMR is a local technique and furthermore, they only see a broadening, not any particular shape of the NMR line. Therefore, they cannot exclude that strain increases primarily the amplitude of the CDW. The authors should be more precise and more careful on this point, especially as they see a splitting at lower doping. Thus, I am tempted to conclude that the CDW under strain is not as long ranged as the CDW at lower doping in high fields.

We would like to emphasize that, in addition to the broadening of the spectral linewidth, we also observed a peak in $1/T_1T$. The "strain-induced" peak in the temperature dependence above 0.15% strain and its presence at $T = 10$ K are evidence of a strain-induced second-order phase transition. Indeed, this behavior is the same as that of the field-induced long range CDW order.

Although we did not observe splitting of the spectrum, it is important to note that the spectrum is broadened isotropically (basically can be fit by gaussian function). This means that the origin of the broadening is a spatially uniform and static charge distribution.

In addition to the linewidth of the satellite peak, we have measured and present the strain dependence of the Knight shift at $T = 10$ K as new evidence in the revised manuscript (revised Fig. 5a-c). The new key evidence supporting a strain-induced long-range CDW order (second-order phase transition) is the combined observation of a decreasing Knight shift and a continuously increasing linewidth above 0.15% strain at $T = 10$ K, where $1/T_1T$ peaks. This was confirmed through careful measurements of the ^{63}Cu -NMR center line under strain. The decrease in the Knight shift indicates a reduced density of states at the Fermi surface, while the increase in linewidth suggests a spatial distribution of the hole concentration. These results indicate continuous growth of the order parameter with respect to strain below T_{CDW} .

Assuming that the origin of the linewidth broadening of the spectrum is the same 1D incommensurate CDW as the field-induced long-range CDW order, the analysis of the spectra shows that the spatial distribution of holes is $\delta p = 0.035$ (Supplementary Fig. 5 and Note 5). This is about half of that of the field-induced CDW order with spectral splitting, which is a reasonable value considering that the T_{CDW} is about half of the field induced CDW order.

3) The following sentence is problematic in several respects, especially within the abstract: “the findings reveal that CDW is an underlying hidden order in the pseudogap state, not limited to the underdoped regime where it is magnetically induced, becoming apparent upon symmetry breaking of the CuO₂ plane by strain”.

- CDW is no longer hidden since it is seen by many probes, including in the studied compound by x-rays and STM. So, from this standpoint the present study “reveals” nothing.

- Also, STM studies in Bi-based cuprates already reported the CDW to persist in overdoped samples, actually throughout the pseudogap regime.

- As far as I know, the origin of the cuprate CDW is still a controversial issue so I am wondering what leads the authors to assert that it is “magnetically induced” in the underdoped regime.

- To me, the intriguing and interesting outcome of this work is the effectiveness of a slight orthorhombic distortion to stabilize strong CDW order. This is what the authors should focus on.

We believe that these concerns have been addressed in the revised manuscript by clearly distinguishing between the “short-range CDW order” observed by X-ray and/or STM measurements and the “long-range CDW order” observed in the presence of a magnetic field or strain, as shown in this paper.

4) Abstract:

- “this sheds light on the physics of cuprates”: rather than such a vague sentence, one would like to know precisely what the results tell us on the physics of cuprates.

- “and may have implications for other strongly correlated electron systems”: this kind of general, unprecise statement should be avoided in my opinion.

We rephrased the sentence as “Our result sheds light on the intertwining of various orders in the cuprates.”.

- Also, on page 11, the authors write “The present results (...) demonstrates that uniaxial strain is a promising tool to elucidate the intertwined orders in strongly correlated electron superconductors”. The verb “demonstrate” may sound slightly over-emphatic. Uniaxial strain has

become a well-established technique in the field of correlated electron systems, with numerous published studies showcasing its effectiveness and yielding remarkable results. In fact, the present results show that the technique has the potential to contribute better understanding of the cuprates, which is already a most remarkable achievement.

We rephrased the sentence as “The present results highlight the importance of the CuO_2 plane symmetry in the cuprates and suggest that uniaxial strain could be a valuable tool to further elucidate the intertwined orders in strongly correlated electron superconductors.”.

5) The whole story about the DOS is unclear and unconvincing:

- it requires to dive into Ref. 12 to understand the normalization factor 0.5 and the notion of relative DOS (that is not defined in the text). Actually, I don't find this relative DOS to be a useful notion here.

- it is far from being proven, and even quite doubtful, that T_1 values reflect the DOS. Due to antiferromagnetic fluctuations, the relaxation rate does not follow the Korringa law of Fermi liquids and furthermore there is quadrupole contribution to T_1 .

- I am not sure how to make sense of a comparison of T_1 values at $T_c(\epsilon)$, i.e. at different temperatures.

We agree with the Reviewer's comment that T_1 does not directly affect the density of states. Therefore, we have performed additional experiments to obtain strain dependence of the Knight shift at $T = 10$ K, which is more directly related to the density of states than T_1 , in order to further support our claims.

In the revised manuscript, we have replaced the strain dependence of relative DOS estimated from $1/T_1T$ (previously Fig. 4c) with the strain dependence of the Knight shift as revised Fig. 5b, based on the results of additional experiments. We have also added a description of the method used to estimate the relative DOS to the Methods section. The strain dependence of the obtained relative DOS was roughly consistent with the values originally obtained from $1/T_1T$, but we have removed the previous Fig. 4c from the paper since the results obtained from the Knight shift are more direct.

- I find it very difficult to be convinced that this data provides any evidence of Fermi surface reconstruction.

- The authors should rather consider as a reasonable assumption that the Fermi surface gets reconstructed as strain strengthens the CDW (although the issue as to how the CDW correlation length affects the Fermi surface reconstruction is a complicated one).

In the revised manuscript, we emphasize the occurrence of a strain-induced phase transition from a short-range CDW to a long-range CDW order. Based on the results of additional experiments, the strain dependence of the Knight shift clearly demonstrates a decrease in the density of states at the Fermi surface accompanying the phase transition. As the Reviewer pointed out, the origin of Fermi surface reconstruction in cuprates is still under debate, and we cannot address its origin in this paper. However, we propose that our results provide at least some evidence for a reduction in DOS caused by the long-range CDW order, which could be linked to the Fermi surface reconstruction.

6) Some effort in sharpening the content is necessary. There are many repetitions and many things that could be written in a much more compact way. This would greatly improve readability. - On page 7 notably, there are many sentences to say that strain induces stronger CDW order while depressing but not suppressing superconductivity: “a long-range CDW order orders in the CuO₂ plane”, “the order parameter (...) is of charge origin”, “induces a CDW order”.

We deleted the sentences “This is evidence that the order parameter of the strain-induced phase transition is of charge origin.” and “, and provide evidence that strain-driven symmetry breaking induces a CDW order in the optimally doped Bi2201 superconductor.”

- On the same page, the sentence “the bottom panel in Fig. 6a (...)” is useless here as it is discussed in another section.

We deleted the sentence “The bottom panel in Fig. 6a is a schematic illustration of the strain dependence of the CDW in real space (details in the Discussion).”.

- Page 10 “a PDW state is a superconducting state with a spatially modulated electron-pair density and energy gap” and “the superconducting order parameter in a PDW state exhibits spatial variations (...)”.

- Etc.

We deleted the words “exhibits spatial variation and” and “the modulation occurs with”.

7) Page 7. “such changes in the spectrum are perfectly consistent with those observed in (...)”. The authors should be more precise. They refer to studies where a line splitting is seen, while they observe a broadening instead of a splitting here. So, it is not immediately obvious what is

consistent with what.

We replaced the sentence “Such changes in the spectrum are...” with “The observed spectral changes, the satellite line being more prominently affected than the central line, are...”.

8) Page 8. In YBCO, an xray scattering study without strain has already found that the short-range CDW consists of domains of unidirectional CDW: Comin et al. Science 347, 1335 (2015). Also, an NMR study reported that the short-range CDW shows in-plane anisotropy at the local scale: Wu et al. Nature Communications 6, 6438 (2015).

9) “under strain, despite the lack of evidence of long-range order”. This should be slightly reformulated so that readers don’t get the impression that strain fails to induce long-range order in YBCO, which is obviously not true. I think it is sufficient to mention here that studies of the short range CDW in YBCO have also concluded that it is locally unidirectional (above refs. + ref. 50.).

We thank the Reviewer for pointing out the missing citations and his/her suggestions.

We have revised the sentence according to the Reviewer’s suggestions as NMR [45] and RXS [35,59] studies of underdoped YBCO superconductor have also suggested a unidirectional nature of the CDW order.

10) Page 8. “lifts degeneracy of energy levels”. A more specific expression than “energy levels” would probably be preferable. Degeneracy of the orthogonal domains?

We thank the Reviewer’s suggestion. We have replaced the words “energy levels” to “the orthogonal domains”.

11) Page 9. Hydrostatic pressure: the transport paper cited by the authors (ref 51) appears to be in contradiction with Xray scattering data: Souliou et al. PRB 97, 020503(R) (2018). There is actually an NMR study of Vinograd et al. about the hydrostatic pressure effect on the CDW in YBCO that does a great job summarizing this issue with all the relevant references: Phys. Rev. B 100, 094502 (2019).

We really thank the Reviewer for providing an update on the current status of hydrostatic pressure experiments on YBCO. To avoid any potential reader misunderstanding, we have replaced the citation in the relevant sentences with a paper on NMR experiment (Vinograd *et al.*, Phys. Rev. B **100**, 094502 (2019).) that summarizes the situation of the hydrostatic pressure effect

on SC and CDW in YBCO.

12) Fig. 6b is quite difficult to read. A rotation of the (p,T) plane would probably help. However, I am afraid there is too much information to make things really clear here. Are the large blue areas depicting CDW fluctuations necessary? To me this is rather confusing. Furthermore, I am afraid this is misleading as these blue areas depict situations in which quadrupole relaxation is particularly strong. However, my understanding of ref. 37 is that quadrupole relaxation, i.e. CDW fluctuations, are present at essentially all other dopings as well. Also confusing is that the phase diagram ignores STM and Xray works reporting the presence of short-range CDW order throughout the pseudogap regime, including the optimal doping discussed here. These remarks obviously apply to Fig. 1a as well.

We would like to express our sincere gratitude to the Reviewer for his/her deep understanding of our experimental results. First, we have added RXS and STM results as hatched patterns to the phase diagram in the revised Fig. 1a. In addition, we have newly cited the results of STM experiments on Bi2201 in the introduction [32,40-42], in addition to the results of RXS for the short range CDW order.

The Reviewer is correct that the 3D phase diagram is too complex and difficult for general readers to understand. We agree that our conclusions can be conveyed sufficiently even without this figure. Therefore, we have deleted the previous Fig. 6b in the revised manuscript. The previous Fig.6a is now shown as Fig. 4.

13) Page 9: “This demonstrates that the CDW order is accompanied by the Fermi surface reconstruction in cuprates”. I don’t get what demonstrates that CDW order is accompanied by Fermi surface reconstruction. Furthermore, the sentence sounds like a general conclusion applying to all cuprates while the authors can only make statements about Bi2201. Actually, why would the Fermi surface not be reconstructed? If there is well defined CDW order, it has to be reconstructed, it is not really an issue.

As mentioned in our response to his/her Remark 5, our new Knight shift results directly demonstrate a reduction in the DOS, likely linked to Fermi surface changes that shrink (reconstruct) concurrently with the formation of the long-range CDW order. We would like to emphasize that these results unequivocally demonstrate a concurrent decrease in the DOS with the onset of long-range CDW order, at least in the optimally doped Bi2201 superconductor. This finding is significant because it provides a direct link between the CDW order and the electronic structure.

While we agree with the Reviewer that it is generally expected for the Fermi surface to reconstruct upon the formation of long-range CDW order, we have rephrased the sentence in remark to avoid any potential misunderstanding from readers.

14) The authors recognize they have nothing to say about PDW order from the present data. Yet, there is more than one page about PDW. I find this unreasonable. Concerning the prospect of addressing this question with NMR, the authors might notice that this has been done by Vinograd et al. *Nat. Commun.* 12, 3274 (2021) in the context of YBCO.

In response to the Reviewer's Remark 6 and this suggestion, we have removed unnecessary phrases and shortened the paragraph about a PDW order. We also appreciate the Reviewer's pointing out the missing citation in our paper, and we have added citation as ref 65.

15) Page 3: "Resonant x-ray scattering (RXS) experiments suggest that the field-induced CDW order found by NMR (...)". The sentence could suggest that the high-field CDW has been seen only by NMR, which is incorrect: it has been also seen by Xray scattering in YBCO (*Science* 350, 949–952 (2015) 914–915 and 3 or 4 subsequent papers).

We deleted the words "by NMR" and "found by NMR" in the paragraph. And we added two high field Xray studies as refs. 27 & 28.

16) Page 3: "which suggested the necessity of completely suppressing superconductivity for the charge-ordered state to emerge in this compound". This is incorrect. Superconductivity and CDW coexist over some field range in YBCO. See Wu et al. *Nat Commun* 4, 2113 (2013) as well as the phase diagrams in these two papers: Kacamarcik et al. *Phys. Rev. Lett.* 121, 167002 (2018) and LeBoeuf et al. *Nat. Phys.* 9, 79–83 (2013). It is also true in the La214 cuprates.

We have deleted the sentence.

17) Page 9: "The present results suggest that superconductivity can survive even though the Fermi surface is reconstructed and/or the static CDW order is present". Same remark as preceding points concerning the SC – CDW coexistence and about the Fermi surface reconstruction.

In order to avoid any potential misunderstandings and to ensure clarity, we have deleted the sentence in remark from the revised manuscript. We believe that the overall meaning of the paper is still clear and coherent without this sentence.

18) Page 4: CDW in the infinite layer nickelates is apparently quite controversial to say the least. People now think the signal is due to oxygen ordering. There are probably better examples to find.

We are grateful for the Reviewer's providing us with the most up-to-date information on Ni compounds. Considering this new information, we have revised the citation in the relevant section of the manuscript to the layered transition metal dichalcogenides (ref.49).

19) In Fig. 5, I initially wondered why the signal from Cu metal is that broad. After reading the Methods, I understood the signal must come from the strain cell as the authors used silver coils. Is this correct? Better specifying the origin of the Cu-metal signal in the caption.

The Reviewer's understanding is correct. We have added a Supplemental Fig. 3 and Note 3 on the origin of the Cu-metal signal.

20) Methods section: "Three homemade strain cells (#3, #4, and #5) were used in parallel to perform all the strain experiments." I am not entirely sure what is the purpose of this information and what "in parallel" precisely means here.

We used three homemade strain cells in parallel to improve data collection efficiency and reduce experimental errors for T_c measurements at $H = 0$ and 13 T shown in Fig. 2. Therefore, we have rephrased the sentence to clarify their intended use.

Responses to the remarks by Reviewer #2:

This manuscript describes Cu NMR studies of a prototypical high T_c cuprate under uniaxial strain. This material is particularly nice because it has a tetragonal crystal structure, unlike other cuprates that are orthorhombic. By applying strain, the C_4 symmetry of the lattice is broken and the authors find that charge ordering is stabilized, while superconductivity is slightly suppressed. The results are interesting, but there are a number of issues with the manuscript that need to be addressed.

We are grateful that the Reviewer finds our experimental results interesting.

(1) The strain values reported do not take into account thermal contraction. The authors report the strain based on length, L , as measured by capacitance, and the original length, L_0 . The latter, however, will be temperature dependent because of the thermal contraction of the crystal. When the temperature is reduced the crystal will likely be under finite tensile strain even in the absence of any applied force, and that " $\epsilon = 0$ " in Figs. 2, 3, 4, and 6, is not really zero strain. Other publications have used an intrinsic measurement that is sensitive to strain in order to determine the zero strain value of L , or used measurements of the applied stress to infer the strain. These include <https://doi.org/10.1103/PhysRevB.108.205113>, <https://www.nature.com/articles/s41467-018-03377-8>, and <https://www.nature.com/articles/s41586-019-1596-2>. None of these other works are cited by the authors, however, and there is no discussion of how to deal with the issue of determining $\epsilon = 0$ at cryogenic temperatures.

We would like to express our sincere gratitude to the Reviewer for his/her important comments regarding the strain experiment. We also apologize for not having described this important point in the initial version of the manuscript.

In fact, we have taken the precaution of reducing the residual strain caused by thermal expansion by fixing the sample to the cell. To minimize thermal strain in our homemade cell, we designed the titanium parts that hold the sample plate to have similar lengths and placed the sample plate in the center of the cell. This configuration effectively reduced residual strain caused by the thermal expansion of both the sample and the cell during NMR experiments.

Prior to conducting the strain experiments on the optimally doped Bi2201 superconductor in this study, we measured the strain dependence of T_c at $H = 0$ for the underdoped Bi2201 superconductor ($p = 0.114$). The T_c exhibits a significantly greater sensitivity to external strain compared to the optimally doped Bi2201 superconductor (unpublished data). This confirmed that the residual strain was negligible, with zero volts ($V_{\text{piezo}} = 0$) corresponding to zero strain at cryogenic temperatures in our cell (Supplementary Fig. 2 and Note 2). We have also confirmed by T_1 (see revised Fig. 1c) and spectral measurement (Supplementary Fig. 3) that even with the optimally doped Bi2201 sample, the results measured with the $V_{\text{piezo}} = 0$ and the bare sample measured without the cell agree.

Supplementary Figs. 1, 2 and Notes 1, 2 provide additional details about our cell and experimental results, which show negligible residual strain at low temperature. The revised manuscript also includes additional details in the "uniaxial strain" of the Methods section. We have also added the papers on the strain experiment to the references [57,80,81] that the Reviewer suggested.

(2) The manuscript contains several incorrect, vague, or misleading statements.

We thank the Reviewer for his/her careful review and pointing out these inaccuracies and ambiguities. We have revised the manuscript as follows to address the Reviewer's concerns:

For example, the abstract states that strain "deliberately breaks translational and rotational symmetries". In fact, strain breaks a discrete rotational symmetry but translation symmetry is already broken in a crystalline lattice even without strain.

In the abstract, to eliminate the inaccuracy, we have rephrased "deliberately breaks translational and rotational symmetries" to "deliberately breaks the crystal symmetry".

In the introduction, the authors state that URu₂Si₂ and the high T_c cuprates are "examples of background electronic states with broken symmetry", referring to the hidden order and 'pseudogap' states. However, they also state that it is "unknown whether any symmetry is broken" in the latter. These statements are contradictory and confusing.

In the introduction, to eliminate the contradiction, we have rephrased "There are two well-known examples of background electronic states with broken symmetry." to "There are two well-known examples of background electronic states that are often discussed as having broken symmetry."

Furthermore, they state that long-range charge order breaks translation symmetry. Presumably they mean that it breaks discrete translation symmetry of the crystalline lattice, since any solid lattice already breaks translation symmetry.

Following the same approach as in the abstract, we have unified the terminology related to symmetry in the main text by using the expression "crystal symmetry of the CuO₂ plane" throughout.

It is also unclear why they state that the triple Q charge order in CsV₃Sb₅ breaks time reversal symmetry.

We deleted the related words.

In the discussion, they state that "high magnetic field and uniaxial strain are effective in breaking new ground in physics of strongly correlated electron systems". This statement seems vague and unnecessary.

We deleted the sentence.

(3) There is no discussion of local strain by dopant atoms. They state that strain is applied along the Cu-O bond direction, "which breaks the local crystal symmetry of the CuO₂ plane". Since the strain field is presumably long range, it is not clear why the symmetry breaking is local. They also refer to "artificial local symmetry breaking".

We apologize that the term "local crystal symmetry" was misleading and inaccurate. In fact, as shown in Fig. 1e, Hooke's law is valid and the stress applied to the single crystal plate is uniform throughout the sample. To clarify this point, we have removed the word "local".

Local symmetry breaking can be caused by the dopants themselves, leading to a random strain field. This leads to an important point: how large is the local strain field from the dopants, versus the externally applied strain field? The authors point out that they see no change in the EFG asymmetry parameter under strain. But couldn't the changes they observe in the quadrupolar satellites reflect such a change? In other words, what would happen to the EFG parameters in the presence of the charge order that the authors hypothesize exists? Wouldn't it naturally give rise to a non-zero η ? Presumably the spectra are broad because of a distribution of local strains and EFGs, and the external strain field only slightly modifies this distribution.

Here, we present the same data as in Fig. 6, comparing the spectra at zero strain ($V_{\text{piezo}} = 0$) and the maximum compressive strain of -0.225% at $T = 28$ K (normal state). As a rough estimate, the asymmetry parameter will be at most $\eta = 0.01$ at a maximum strain of -0.225% (-0.00225) in this experiment. With such a small asymmetry parameter, the NQR frequency ν_Q hardly changes, and in the case of Bi2201, it only changes from $\nu_Q = 28.4$ to 28.4005 MHz ($\nu_Q = \frac{3eQV_{zz}}{2I(2I-1)h} \sqrt{1 + \eta^2/3}$ [71]). Obviously, our spectrum does not have such high precision. In other words, the observed line broadening cannot be explained by the effect of η due to external strain. We added the above formula in the Method section.

We would like to emphasize that the observed linewidth broadening emerges at low temperatures above the critical strain of 0.15 %, notably accompanying the peak of $1/T_1T$. This suggests a distinct mechanism, as a temperature-independent local strains possibly arising from dopants

cannot explain this behavior. Therefore, the most natural interpretation is that it originates from the spatial distribution of holes, namely the formation of CDW below $T_{\text{CDW}}(\epsilon)$.

(4) They point out that the CDW order in the Bi2201 sample is insufficient to lead to peak splitting in their NMR spectra "due to the lower T_{CDW} ". Presumably they are assuming that the magnitude of the CDW order parameter is proportional to T_{CDW} when they make such a statement, but they should clarify this point.

To check the validity of our hypothesis, we performed an additional analysis of the spectrum assuming a splitting due to CDW formation, similar to the case of magnetic field-induced long-range CDW order. The results are shown in the Supplementary Fig. 5 and Note 5. The analysis revealed that the magnitude of the spatial distribution of the holes (CDW order parameter) and the T_{CDW} in the CDW ordered state of the strain-induced long-range CDW order are indeed half of those of the magnetic field-induced long-range CDW order, indicating a correlation between the order parameter and T_{CDW} . This correlation suggests that the strain-induced long-range CDW order and the magnetic field-induced long-range CDW order may share a common underlying mechanism.

These results suggest that applying further strain, if possible, could lead to an increase in T_{CDW} , which would be highly promising for observing spectral splitting. We plan to explore this possibility in future studies.

(5) The authors need to address their results in the context of other NMR studies of cuprates under strain. How does one reconcile the fact that strain seems to induce static charge order in Bi2201, but has essentially no effect on charge order in the 214 system (see [10.1103/PhysRevB.108.205113](https://arxiv.org/abs/10.1103/PhysRevB.108.205113))?

We thank the Reviewer for introducing recent strain study on La214 system. We added the

sentences “The present results also contrast with recent findings that there is essentially no strain effect on the charge order in underdoped $\text{La}_{1.875}\text{Ba}_{0.125}\text{CuO}_4$. ” just before the Discussion and added the paper as a reference [57].

REVIEWER COMMENTS

Reviewer #1 (Remarks to the Author):

The authors have essentially taken all my comments into account. However, there remains a number of problematic points that need to be addressed in the manuscript before publication:

1. Page 7: "likely reflecting a spatial variation in the hole concentration within the CuO₂ planes." Although this sentence is not incorrect, it may sound to some readers like a distribution of doping. I presume that the authors have a charge density wave (CDW) in mind here, in which case something like "variation in the carrier density" would probably be less ambiguous.

2. I am surprised by the amplitude of the charge modulation, which seems to be quite large (especially in the field-induced phase), larger than in YBCO or in La₂14 cuprates if I am not mistaken. Could the authors comment on this?

3. The pictures in Fig. 1d are too small to discern anything. The authors should either shift (an enlarged version of) this figure to the supplementary part or construct Fig. 1 in a different way to enlarge the different panels.

4. Why is the zero-field T_c strain insensitive whereas the T_c under the field is strain-sensitive? Could it be that the claimed long-range order arises from the combination of strain and magnetic field? If so, shouldn't this be probed by NQR?

5. Fig. 6 suggests that +/- 0.2% deformation does not change the width of the central line, which would contradict Fig. 5c. I suppose the small change reported in Fig. 5c is barely visible in Fig. 6, but this might be a bit confusing for the readers. Changing the scale in Fig. 6d to zoom in on the data below the axis break could mitigate the risk of confusion.

6. The evidence that the CDW order is long-range is quite indirect, and I think the authors should acknowledge this more clearly and describe the different steps of their reasoning. 1) There cannot be any direct evidence from NMR because of the local nature of this probe. 2) There is no evidence of a phase transition from the spectra, and given the width of NMR lines in Bi₂201, there can hardly be any hint as to whether the CDW correlation length gets long. 3) However, there is evidence of a phase transition from T₁ measurements. It is then natural to link the apparent modification in the CDW pattern to the phase transition, which inevitably leads to the proposal that the CDW undergoes long-range ordering. Can the authors exclude that the transition is an incommensurate-to-commensurate transition? Has this ever been seen to produce a peak in T₁ data?

7. By the way, most second-order phase transitions lead to a divergence in 1/T₁ vs. T. Here, we only have a (admittedly substantial) cusp in 1/T₁ vs. T. Is the effect strong enough to unambiguously prove that it is a phase transition? How does 1/T₁ vs. T data look like?

Reviewer #2 (Remarks to the Author):

The authors have satisfactorily addressed my concerns, and the manuscript is stronger now with the inclusion of the new Knight shift results. It is still surprising to me that there is no residual strain at low temperatures, because it is highly unlikely that both the sample and the cell would have the same thermal contraction. Nevertheless, the data shown in supplementary figure 2 makes a compelling argument that V_{piezo} corresponds to zero strain.

One point that should be addressed is how they determined the stress (labeled as Pressure) on the upper axis of Fig. 4. Of course this quantity should be proportional to strain, which they measure with the strain cell capacitor. Do they also have a measure of the stress, or if they are using some sort of conversion factor, how is this determined?

Other than this, I recommend publication.

RESPONSES to Reviewers' remarks

We would like to express our sincere gratitude to the Reviewers for their time again. Their additional comments and constructive suggestions have further improved the quality of our manuscript.

We have carefully considered their feedback and have made the following revisions:

Responses to the remarks by Reviewer #1:

The authors have essentially taken all my comments into account. However, there remains a number of problematic points that need to be addressed in the manuscript before publication:

Thank you for the additional comments.

We have carefully considered the reviewer's comments and have made some changes to the manuscript in an effort to improve its clarity and comprehensiveness.

1. Page 7: "likely reflecting a spatial variation in the hole concentration within the CuO₂ planes." Although this sentence is not incorrect, it may sound to some readers like a distribution of doping. I presume that the authors have a charge density wave (CDW) in mind here, in which case something like "variation in the carrier density" would probably be less ambiguous.

We appreciate the reviewer's suggestion. We have revised the sentence in remark as suggested by the reviewer.

2. I am surprised by the amplitude of the charge modulation, which seems to be quite large (especially in the field-induced phase), larger than in YBCO or in La₂14 cuprates if I am not mistaken. Could the authors comment on this?

Regarding the reviewer's concern about the difference in the amplitude of the spatial hole distribution between Bi2201 and YBCO, we have previously discussed this issue in our paper (ref. 29). In that paper, we proposed that the difference is due to the difference in the number of CuO₂ planes between the two systems. YBCO is a bi-layer system, while Bi2201 is a single-layer system. We hypothesized that when CDW has a different phase between the two CuO₂ planes in YBCO, the ordering effect would be weakened or even canceled out.

3. The pictures in Fig. 1d are too small to discern anything. The authors should either shift (an enlarged version of) this figure to the supplementary part or construct Fig. 1 in a different way to enlarge the different panels.

We appreciate the reviewer's recommendation. We have moved Fig. 1d and 1e on the strain experimental technique to the supplementary information as Supplementary Fig. 1 and Fig. 2, respectively.

4. Why is the zero-field T_c strain insensitive whereas the T_c under the field is strain-sensitive? Could it be that the claimed long-range order arises from the combination of strain and magnetic field? If so, shouldn't this be probed by NQR?

We appreciate the reviewer's important question, which goes to the heart of the CDW in cuprates. To address the reviewer's question, the insensitivity of T_c to strain at zero field in the optimally doped sample remains unclear. While our current data hint at possible explanation, these are not yet supported by conclusive evidence for inclusion in the main text.

Our preliminary experiments on the underdoped Bi2201 superconductor $p = 0.114$ showed that T_c can be suppressed by strain even at zero magnetic field (Supplementary Fig. 3). Similarly, the underdoped YBCO ($p = 0.12$), despite being anisotropic response, also exhibits T_c suppression by strain under zero magnetic field (refs. 54 & 55). A common feature of these two samples is the emergence of field-induced long-range CDW order at magnetic fields above 10 T without strain (refs. 26 & 29). On the other hand, the optimally doped Bi2201 superconductor does not exhibit magnetic field-induced long-range CDW order even under a 45 T field. Therefore, Fig. 2 aligns with the reviewer's concern, suggesting that CDW in Bi2201 can be enhanced by both strain and field.

Verifying this hypothesis requires further investigation of the magnetic field, strain, and doping dependence of CDW and superconductivity. However, it is clear that the work is extremely challenging and deviates from the main focus of the present study. Therefore, we will leave it as a future work.

The reviewer rightly suggests NQR experiments under strain as a key tool. However, the NQR signal is often too weak in single crystal plates typically used for strain experiments.

We acknowledge that this is an important issue that needs to be addressed in the future.

5. Fig. 6 suggests that +/- 0.2% deformation does not change the width of the central line, which would contradict Fig. 5c. I suppose the small change reported in Fig. 5c is barely visible in Fig. 6, but this might be a bit confusing for the readers. Changing the scale in Fig. 6d to zoom in on

the data below the axis break could mitigate the risk of confusion.

We appreciate the reviewer's suggestion. Following the suggestion, we have changed the vertical axis scale of the ^{63}Cu center in Fig. 6d to be the same as that in Fig. 5c. We believe that this change makes it easier to see the increase in the linewidth of the center peak.

6. The evidence that the CDW order is long-range is quite indirect, and I think the authors should acknowledge this more clearly and describe the different steps of their reasoning. 1) There cannot be any direct evidence from NMR because of the local nature of this probe.

Thank you for pointing out the need for a clearer discussion on the evidence for long-range CDW order in our study. We completely agree that the current description could benefit from a more nuanced explanation.

Bi2201 contains only one Cu site within the single CuO_2 plane, and Cu plays a central role. Given that the physics of cuprates is primarily based on the CuO_2 plane, it is natural to assume that our Cu-NMR adequately captures the electronic state of Bi2201 from a bulk perspective. It is also important to note that NMR being a local probe does not imply that it cannot provide direct evidence for CDW. This is evident from the numerous NMR experiments on CDW, including our own work (ref. 29 & 75), that have been reported in the past for cuprates and other CDW materials.

It is significant that NMR experiments played a crucial role in the discovery of magnetic field-induced long-range CDW order in cuprates (ref. 26).

2) There is no evidence of a phase transition from the spectra, and given the width of NMR lines in Bi2201, there can hardly be any hint as to whether the CDW correlation length gets long. 3) However, there is evidence of a phase transition from T1 measurements. It is then natural to link the apparent modification in the CDW pattern to the phase transition, which inevitably leads to the proposal that the CDW undergoes long-range ordering.

We acknowledge that our previous explanation in the revised manuscript may have been unclear. As Reviewer #2 acknowledges, "*the manuscript is stronger now with the inclusion of the new Knight shift results*". The strain dependence of the NMR center spectrum, which we added in the revised manuscript, is important because it provides evidence of long-range order (second-order phase transition).

As shown in Fig. 5a and b, the NMR spectrum shifts to lower frequency (K_s decreases) above the critical strain, keeping its Gaussian shape. This suggests that the DOS decreases uniformly at all Cu sites, indicating a uniform change in the shape of the Fermi surface.

Furthermore, we have realized that the strain dependence of the FWHM (Fig. 5c) follows a mean-field behavior $(\varepsilon - \varepsilon_c)^{0.5}$ above the CDW phase boundary ($\varepsilon_c = -0.152\%$) at 10 K (see Fig. 4). We have added dashed curve in Fig. 5c.

These results provide clear microscopic evidence for a long-range order (second-order phase transition) at ε_c , characterized by a uniform growth of the order parameter [charge (hole) distribution amplitude] within the CuO_2 plane.

Notably, the mean-field behavior of the charge distribution amplitude is consistent with that observed for the magnetic field-induced long-range CDW order in YBCO [Fig. 1f in ref. 26 and Fig. 3 in Nat. Commun. 4:2113 doi: 10.1038/ncomms3113 (2013).] and Bi2201 (Fig. 4 in ref. 29), and we believe it provides compelling evidence for a long-range CDW order under strain.

Therefore, contrary to the reviewer comments, our T_1 and NMR spectrum experimental results support the conclusion of a strain-induced long-range CDW order (second-order phase transition). The only unknown aspect remains the details of the CDW nature (dimensionality, stripe, checkerboard type) because no characteristic structural features appeared in the NMR spectra. Therefore, we compare with the results of X-ray at zero strain.

In the revised manuscript, detailed explanations have been added to the paragraphs pertaining to Figures 5b, 5c, and 5d in the section “Strain dependence of the phase transition at $T = 10\text{ K}$ ”.

Can the authors exclude that the transition is an incommensurate-to-incommensurate transition? Has this ever been seen to produce a peak in T_1 data?

If the reviewer is referring to the “incommensurate-to-incommensurate transition” for short-range CDW, which is not a phase transition in the strict sense, then there would be no peak in the temperature dependence of $1/T_1$, and the concern can be completely ruled out. Even if the transition is from one CDW state to another, as the reviewer seems to suggest, only a second-order phase transition between distinct phases can generally exhibit a peak in the temperature dependence of $1/T_1$.

While the CDW nature may change upon the short-range to long-range order transition, our study does not provide conclusive evidence on this matter. However, we believe this is a mere detail for the strain-induced second-order phase transition from short-range to long-range order observed in our study, and it will be clarified in future.

7. By the way, most second-order phase transitions lead to a divergence in $1/T_1$ vs. T . Here, we only have a (admittedly substantial) cusp in $1/T_1$ vs. T . Is the effect strong enough to unambiguously prove that it is a phase transition? How does $1/T_1$ vs. T data look like?

In our experience, the magnitude of the jump in $1/T_1T$ should not be a major concern. The magnitude of the peak can depend on the environment around the observed nucleus, such as the magnitude and anisotropy of the hyperfine coupling constant and/or susceptibility.

While the reviewer may find the jump in $1/T_1T$ to be small, it is important to note that the size of the jump should be considered in the context of the long-range CDW order appearing in a pseudogap background (see Figure **a** below). At $\varepsilon = -0.225\%$, although it is a rough approximation, if we assume that the jump appears as a linear decrease of $1/T_1T$ due to the pseudogap, as shown in the inset of the Figure **a** below, the size of the jump at $T_{\text{CDW}} = 28$ K is estimated to be about 1.3 times that of $1/T_1T$ in the normal state. Although, a direct comparison is difficult, our previous experiment on SrPt_2As_2 , a material that exhibits two 1D incommensurate long-range CDW orders (similar to Bi2201 's long range CDW order), showed jumps of approximately 1.5-1.6 times at $T_{\text{CDW}} = 410$ K and 255 K, respectively (Fig. 3 in ref. 75). Generally speaking, considering the difference in transition temperature by an order of magnitude, the jump itself in Bi2201 's CDW order is not small.

We believe this effect is strong enough to definitively prove a phase transition, as evidenced by the accompanying peak in $1/T_1T$ alongside the decrease in Knight shift and increase in linewidth (indicative of a growing static order parameter).

As shown in Figure **b** below, the anomaly at T_{CDW} is clearly visible even in the $1/T_1$ vs T plot.

it would be nice if the authors could add error bars to their Knight shift and T1 measurements. If these are smaller than the symbol size, this should be specified.

We have added error bars to Fig. 3a, b and 5b, c.

Responses to the remarks by Reviewer #2:

The authors have satisfactorily addressed my concerns, and the manuscript is stronger now with the inclusion of the new Knight shift results.

We thank the reviewer for his/her time reviewing our revised manuscript and we are happy that the changes have “satisfactorily addressed his/her concerns, and the manuscript is stronger now with the inclusion of the new Knight shift results”.

It is still surprising to me that there is no residual strain at low temperatures, because it is highly unlikely that both the sample and the cell would have the same thermal contraction. Nevertheless, the data shown in supplementary figure 2 makes an compelling argument that V_{piezo} corresponds to zero strain.

We appreciate the reviewer’s comment. We agree that the thermal expansion of the strain cell and the sample are not identical. We believe that Supplementary Fig. 1 & 3 demonstrate that our cell design was able to minimize the influence of low-temperature residual strain on the strain response of the Bi2201 electronic state in this experiment. It is clear that it is very important to check for residual strain for each compound in future strain experiments, and we hope that our strain experiments will be a useful reference for others.

One point that should be addressed is how they determined the stress (labeled as Pressure) on the upper axis of Fig. 4. Of course this quantity should be proportional to strain, which they measure with the strain cell capacitor. Do they also have a measure of the stress, or if they are using some sort of conversion factor, how is this determined?

We apologize for any confusion caused by our ambiguous wording. Our strain cell does not have a force sensor, so we did not directly determine the pressure value in addition to the strain. The pressure value was derived from the strain value using the elastic constants obtained from

the first-principles calculation in ref. 82. Since it is not a directly measured value, it does not have quantitative accuracy. However, we considered that pressure is more familiar to general readers than strain, so we provided it as a reference value. Additionally, we noticed that the paper (ref. 57) introduced by the reviewer in his/her first remark also determines the stress in LBCO in a similar manner to ours, albeit using experimentally measured elastic constants. We are happy to see the relationship between stress and strain in their work is very similar to our values, suggesting that our estimated pressure values are not unrealistic. Therefore, we believe that they are meaningful values as reference values.

Nevertheless, we have revised the sentence in the Methods section to "Our cell does not have a force sensor to directly measure pressure alongside strain. Therefore, for reference, we estimated pressure values using the elastic constants of $\text{Bi}_2\text{Sr}_2\text{CuO}_6$ from prior first-principles calculations [82]. These estimated values are shown on the upper axis of Fig. 4. Of note, our value is in good agreement with that reported in strain experiments on the related cuprate $\text{La}_{1.875}\text{Ba}_{0.125}\text{CuO}_4$ superconductor [57]." to avoid misleading readers.

Other than this, I recommend publication.

We believe the manuscript is significantly improved due to the reviewers' insightful comments. We are grateful for their feedback.

REVIEWERS' COMMENTS

Reviewer #1 (Remarks to the Author):

I'm not 100% convinced by some of the authors' answers, but these are only minor details, and my role as referee is certainly not to prolong this discussion. For its overall quality and interest in the subject, this paper amply deserves to be published in Nature Communications.

Reviewer #2 (Remarks to the Author):

The authors have satisfactorily addressed the concerns raised, and the manuscript is improved. I recommend publication.